# Mitochondrial defects caused by PARL deficiency lead to arrested spermatogenesis and ferroptosis

Enrico Radaelli[1], Charles-Antoine Assenmacher[1], Jillian Verrelle[1], Esha Banerjee[1], Florence Manero[2], Salim Khiati[3], Anais Girona[3], Guillermo Lopez-Lluch[4,5], Placido Navas[4,5], Marco Spinazzi[3,6]*

[1]Department of Pathobiology, Comparative Pathology Core, School of Veterinary Medicine, University of Pennsylvania, Philadelphia, United States; [2]University of Angers, Angers, France; [3]Unité Mixte de Recherche (UMR) MITOVASC, Centre National de la Recherche Scientifique (CNRS) 6015, Institut National de la Santé et de la Recherche Médicale (INSERM) U1083, University of Angers, Angers, France; [4]Centro Andaluz de Biología del Desarrollo, Universidad Pablo de Olavide-Consejo Superior de Investigaciones Científicas-Junta de Andalucía, Sevilla, Spain; [5]CIBERER, Instituto de Salud Carlos III, Madrid, Spain; [6]Neuromuscular Reference Center, Department of Neurology, CHU Angers, Angers, France

*For correspondence: marco.spinazzi@chu-angers.fr

Competing interest: The authors declare that no competing interests exist.

**Abstract** Impaired spermatogenesis and male infertility are common manifestations associated with mitochondrial diseases, yet the underlying mechanisms linking these conditions remain elusive. In this study, we demonstrate that mice deficient for the mitochondrial intra-membrane rhomboid protease PARL, a recently reported model of the mitochondrial encephalopathy Leigh syndrome, develop early testicular atrophy caused by a complete arrest of spermatogenesis during meiotic prophase I, followed by degeneration and death of arrested spermatocytes. This process is independent of neurodegeneration. Interestingly, genetic modifications of PINK1, PGAM5, and TTC19 – three major substrates of PARL with important roles in mitochondrial homeostasis – fail to reproduce or modify this severe phenotype, indicating that the spermatogenic arrest arises from distinct molecular pathways. We further observed severe abnormalities in mitochondrial ultrastructure in PARL-deficient spermatocytes, along with prominent electron transfer chain defects, disrupted coenzyme Q (CoQ) biosynthesis, and metabolic rewiring. These mitochondrial defects are associated with a germ cell-specific decrease in GPX4 expression leading arrested spermatocytes to ferroptosis – a regulated cell death modality characterized by uncontrolled lipid peroxidation. Our results suggest that mitochondrial defects induced by PARL depletion act as an initiating trigger for ferroptosis in primary spermatocytes through simultaneous effects on GPX4 and CoQ – two major inhibitors of ferroptosis. These findings shed new light on the potential role of ferroptosis in the pathogenesis of mitochondrial diseases and male infertility warranting further investigation.

## Editor's evaluation

This manuscript reports an important finding that spermatogenic defects in Parl KO mice, a genetic model for Leigh syndrome, may result from mitochondrial defects leading to ferroptosis. The finding is of significance because male germ cell ferroptosis has not been well characterized before. The data as a whole strongly support ferroptosis as a mechanism for germ cell death in the Parl KO. However, potential non-ferroptosis and 'accidental' necrosis cannot be excluded, and the potential effects of quantitative immunofluorescent staining, instead of assays using purified spermatogenic cells, on the conclusion drawn should be considered.

**eLife digest** Up to 9% of men are thought to experience infertility. These individuals may not produce enough healthy sperm cells. The root cause of infertility is often not discovered but, in some cases, it is associated with genetic defects in cell compartments known as mitochondria.

Mitochondria are responsible for converting energy from food into a form of chemical energy cells need to power vital processes. However, it remains unclear how defects in mitochondria contribute to male infertility.

Leigh syndrome is one of the most prevalent and severe diseases caused by genetic defects in mitochondria. The condition often develops in childhood and affects the nervous system, muscle and other organs, leading to many symptoms including muscle weakness and neurological regression. A previous study found that mutant mice that lack an enzyme, called PARL, display symptoms that are similar to those observed in humans with Leigh syndrome. PARL is found inside mitochondria where it cuts specific proteins to ensure they are working correctly in the cells.

Radaelli et al. used extensive microscopy and biochemical analyses to study the fertility of male mice lacking PARL. The experiments revealed that the males were infertile due to a failure to produce sperm: spermatocytes, which usually develop into sperm cells, where much more likely to die in mice without PARL (by a process known as ferroptosis).

Further experiments demonstrated that the mitochondria of the mutant mice had a shortage of two crucial molecules, a protein called GPX4 and a lipid called Coenzyme Q, which are required to prevent death by ferroptosis. It appears that this shortage was responsible for the demise of spermatocytes in the male mutant mice affected by infertility.

These findings reveal a new role for PARL in the body and provide evidence that mitochondrial defects in living mammals can trigger ferroptosis, thereby contributing to male infertility. In the future, this research may pave the way for new treatments for male infertility and other diseases associated with defects in mitochondria.

## Introduction

Impaired spermatogenesis and consequent infertility are increasingly prevalent medical concerns affecting approximately 9% of the global male population (*Boivin et al., 2009*). The underlying mechanisms of these conditions appear to involve oxidative stress and mitochondrial dysfunction, but their specific contribution is poorly characterized (*Aitken et al., 2022*). Furthermore, male infertility has been identified as a significant manifestation of mitochondrial diseases (*Martikainen et al., 2017*). While the essential roles of mitochondria in reproductive biology, including spermatogenesis, are established, their precise mechanisms remain incompletely understood (*Cannon et al., 2011*; *Rajender et al., 2010*). Mitochondrial diseases encompass a range of inborn errors of metabolism caused by genetic defects in either mitochondrial or nuclear genome. The selective vulnerability of specific organs or tissues to these genetic defects remains an enigma and is likely influenced by cell-type-specific activation of poorly understood downstream molecular pathways acting independently of or in parallel with mitochondrial respiratory chain defects. Notably, energy insufficiency alone cannot fully explain the extremely heterogenous clinical manifestations observed (*Dogan et al., 2014*). Consequently, complex molecular responses to mitochondrial dysfunction are gaining recognition as crucial pathogenetic mechanisms (*Suomalainen and Battersby, 2018*; *Khan et al., 2017*; *Forsström et al., 2019*).

In our previous study, we described PARL-deficient mice as a novel model of mitochondrial encephalopathy resembling Leigh syndrome (*Spinazzi et al., 2019*), one of the most common and severe mitochondrial diseases. PARL, an evolutionary conserved protease belonging to the rhomboid family, is located in the inner mitochondrial membrane and has fundamental roles in cell homeostasis. PARL has been associated with various human disorders such as Parkinson's disease, Leber hereditary optic neuropathy, and type 2 diabetes, albeit with some controversy (*Shi et al., 2011*; *Hatunic et al., 2009*; *Istikharah et al., 2013*; *Spinazzi and De Strooper, 2016*). Notably, PARL's significant role in maintaining mitochondrial fitness has been established through critical studies that identified its substrates, such as PINK1 (*Jin et al., 2010*), a mitochondrial kinase implicated in Parkinson's disease and mitophagy (*Valente et al., 2004*; *Yan et al., 2020*), PGAM5 (*Sekine et al., 2012*), a mitochondrial

phosphatase implicated in Parkinsonism in mice (*Lu et al., 2014*), and TTC19 (*Saita et al., 2017*), a mitochondrial protein involved in maintaining complex III activity and associated with human Leigh syndrome (*Bottani et al., 2017*; *Atwal, 2014*).

In this study, we focus on impaired spermatogenesis as the earliest phenotype observed in PARL-deficient male mice, which occurs independently of neurodegeneration. We find that PARL deficiency leads to severe functional and structural abnormalities in germ cell mitochondria, resulting in a complete arrest of spermatogenesis and triggering ferroptosis specifically in spermatocytes. Our findings offer new insights into the role of mitochondrial dysfunction and ferroptosis in male infertility and pave the way for further investigations on this cell death mechanism in mitochondrial diseases.

## Results
### PARL deficiency results in arrested spermatogenesis and severe testis atrophy

PARL-deficient mice appear clinically normal until around 6 weeks of age, after which they develop a progressive necrotizing encephalomyelopathy resembling Leigh syndrome with death before the age of 8 weeks (*Spinazzi et al., 2019*). As previously described, these mice exhibit severe testis atrophy (*Figure 1A*; *Spinazzi et al., 2019*; *Cipolat et al., 2006*). We did not observe cryptorchidism. Upon closer examination, the testis weight of $Parl^{-/-}$ mice at 5 weeks of age, when they do not show clinical signs of neurological impairment, is found to be nearly half of that in matched WT littermates (*Figure 1A*). This difference cannot be explained by concurrent body weight reduction (*Figure 1A*). Microscopic analysis reveals that the seminiferous tubules from $Parl^{-/-}$ mice are smaller in diameter (*Figure 1B*; p=0.0002, *Figure 1—figure supplement 1A*) and contain approximately 40% fewer cells compared to WT littermates (p=0.0009, *Figure 1—figure supplement 1B*). Further investigation indicates that the seminiferous tubules in $Parl^{-/-}$ mice are populated by immature germ cells exhibiting degenerative changes and prominent intraluminal exfoliation, often in the form of multinucleated syncytia (*Figure 1B*). Immunohistochemistry reveals that PARL deficiency leads to a complete meiotic prophase I arrest as the seminiferous tubules are predominantly populated by SCP-1-positive spermatocytes (*Figure 1B*; *Figure 1—figure supplement 1C*; *Yang and Wang, 2009*; p=0.0001) while spermatids and spermatozoa are completely absent (*Figure 1B*; *Köhler, 2007*). The epididymal ducts of $Parl^{-/-}$ mice are also smaller in diameter and completely devoid of mature sperm (*Figure 1B*). Spermatogonia show a modest increase in number in $Parl^{-/-}$ seminiferous tubules compared to WT littermates (p=0.01; *Figure 1—figure supplement 1D*). Additionally, the analysis of γH2AX expression pattern in $Parl^{-/-}$ seminiferous tubules indicates specific meiotic prophase I arrest at the pachytene stage (*Figure 1—figure supplement 1G*). Supporting Sertoli cells appear to be slightly increased in number (p=0.0382; *Figure 1—figure supplement 1E and F*), and the distribution and morphology of Leydig cells appears normal.

To determine whether the observed testicular abnormalities are linked to neurodegeneration, mice with conditional deletion of *Parl* in the nervous system ($Parl^{L/L}::Nes^{Cre}$) were studied. Surprisingly, despite developing severe Leigh-like encephalopathy, these mice exhibit normal testicular size, histology, and sperm production comparable to WT littermates (*Figure 1B*), demonstrating that the testicular disorder is not a consequence of neurodegeneration. As previously reported (*Anand-Ivell et al., 2017*; *Jiang et al., 2014*; *Davidoff et al., 2004*), *Nes* is also expressed in Leydig cells (*Figure 1—figure supplement 2*). Although PARL deficiency *in situ* could not be formally verified in the absence of specific PARL antibodies suitable for immunohistochemistry, Cre recombinase activation under the *Nes* promoter is predicted to effectively delete *Parl* in these cells as in the nervous system. Moreover, extensive morphological observations detailed in the following paragraph indicate that Leydig cells are structurally unaffected in the germline $Parl^{-/-}$ testis (*Figure 2—figure supplement 1B*), suggesting that the spermatogenetic defect is not secondary to PARL deficiency in these cells.

Altogether, deficiency of PARL leads to a complete arrest of spermatogenesis at the level of primary spermatocytes, independent of the effects of PARL in the nervous system and in Leydig cells.

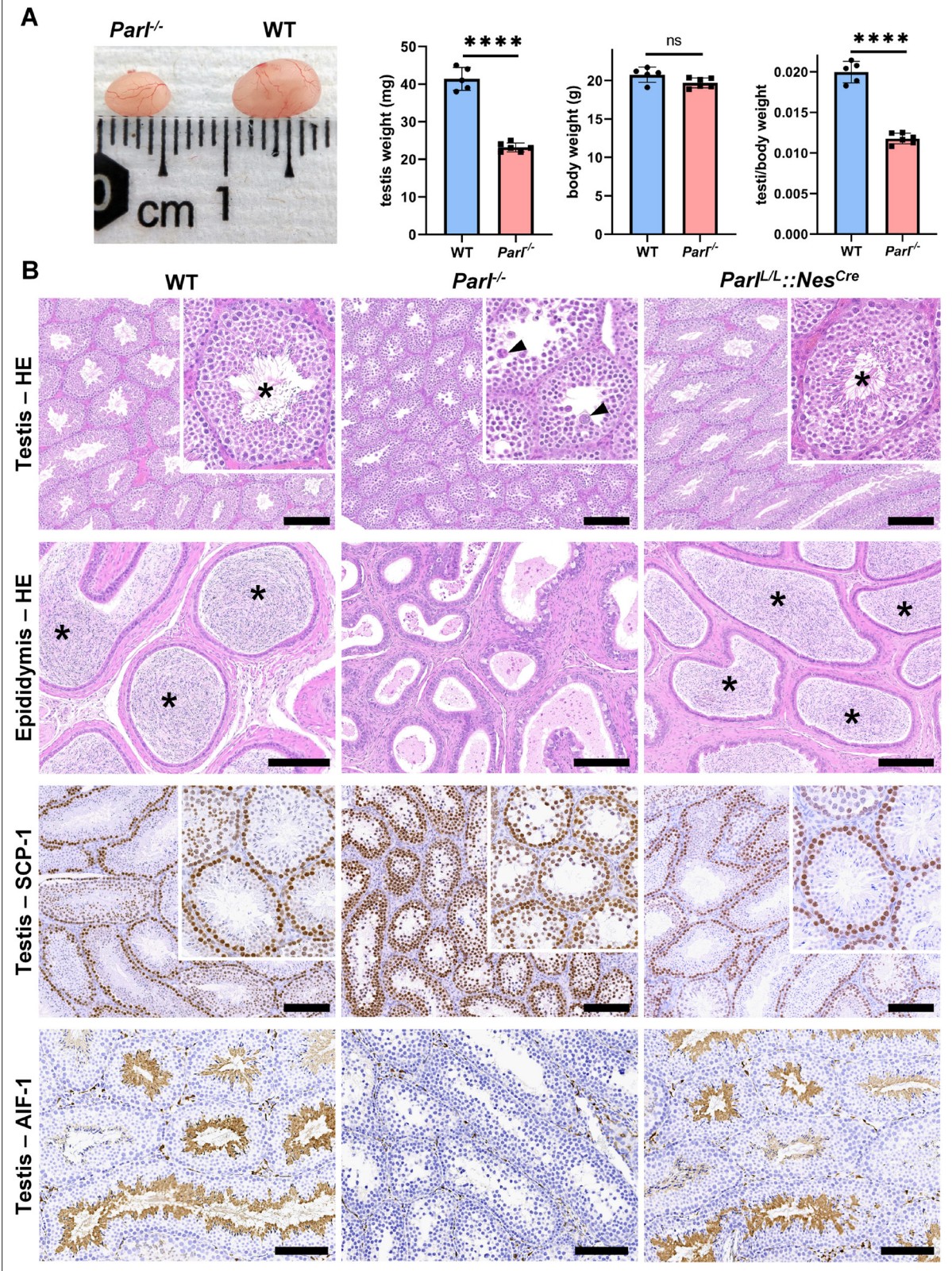

**Figure 1.** Severe testis atrophy in *Parl⁻/⁻* mice is caused by arrested spermatogenesis. (**A**) Reduced testicular size and weight in 5-week-old *Parl⁻/⁻* mice (n = 5) compared to WT littermates (n = 6; unpaired two-tailed *t*-test, p-value<0.0001). The reduction in testicular weight is not explained by body weight differences (p=0.0598). (**B**) Histological assessment of testes from 6-week-old *Parl⁻/⁻* and WT mice reveals reduced diameter of *Parl⁻/⁻* seminiferous tubules with impaired germ cell maturation and complete spermatogenesis arrest at the level of primary spermatocytes (testis HE stain, n = 10 for

*Figure 1 continued on next page*

*Figure 1 continued*

each genotype). *Parl*[-/-] seminiferous tubules also exhibit intraluminal exfoliation of degenerated spermatocytes often in the form of multinucleated syncytia (testis HE stain inset, arrowheads). The complete arrest of spermatogenesis leads to total absence of sperm in *Parl*[-/-] seminiferous tubules and epididymis compared to WT littermates (testis and epididymis HE stain, n = 10 for each genotype; asterisks indicate mature spermatozoa in the WT). Immunohistochemistry for synaptonemal complex protein 1, SCP-1, confirms complete spermatogenesis arrest at the level primary spermatocytes in *Parl*[-/-] testis (testis SCP-1, n = 10 for each genotype). The distribution of SCP-1 expression is confined to primary spermatocytes and is lost in postmeiotic germ cells as they undergo maturation in WT seminiferous tubules. Immunohistochemistry for allograft inflammatory protein 1, AIF-1, reveals the complete absence of spermatids in *Parl*[-/-] testis while WT seminiferous tubules are densely populated by AIF-1-positive spermatids at different levels of maturation (testis AIF-1, n = 10 for each genptype). 8-week-old mice with conditional *Parl* deletion driven by the *Nes* promoter in the nervous system and Leydig cells (*Parl* [L/L]::*Nes*[Cre]) display a normal testicular and epididymal histology as well as SCP-1 and AIF-1 immunohistochemistry comparable to WT mice (right column, n = 4). Scale bars, 200 μm.

The online version of this article includes the following figure supplement(s) for figure 1:

**Figure supplement 1.** Quantitative morphometry, cell composition, and meiotic stage evaluation in 5-week-old WT and *Parl*[-/-] seminiferous tubules.

**Figure supplement 2.** Nestin expression in Leydig cells.

## PARL deficiency results in mitochondrial ultrastructural abnormalities and progressive degeneration and death of arrested spermatocytes

To gain insight into the possible pathological effects of PARL deficiency on germ cells, we conducted a detailed morphological analysis using semithin sections and electron microscopy.

In unaffected WT animals, germ cells undergo a maturation process, with less differentiated forms (spermatogonia and spermatocytes) in the abluminal layers, more differentiated spermatids in the adluminal compartment, and mature spermatozoa in the lumen of the seminiferous tubules (*Figure 2A*). Conversely, *Parl*[-/-] mice exhibit severe vacuolar degeneration of arrested spermatocytes, leading to cell death, and this degeneration progressively worsens from the abluminal to the adluminal compartment (*Figure 2A and B* and *Figure 2—figure supplement 1A*). Analysis of spermatocyte ultrastructure showed a significant increase in the occurrence of degeneration/death in *Parl*[-/-] spermatocytes compared to WT (18.9% degenerated spermatocytes out of 201 analyzed in *Parl*[-/-] vs. 0% out of 79 WT spermatocytes analyzed; n = 3 for each genotype; p=0.0002 by two-sided Fisher's exact test). Next, we assessed whether mitochondrial morphology was affected in PARL-deficient spermatocytes. Differentiation *per se* leads to important morphological adaptations of mitochondria that parallel increasing bioenergetic demands requiring a shift from more glycolytic to more oxidative metabolism (*Varuzhanyan and Chan, 2020*). To ensure accurate comparisons, we focused on primary spermatocytes showing fully assembled synaptonemal complexes, a characteristic feature during the zygotene and pachytene stages of meiotic prophase I (*Figure 2C*; *Yang and Wang, 2009*; *Martins and Silva, 2001*). Compared to the mitochondria of WT primary spermatocytes, which are typically small with dilated cristae and dense finely granular matrix, mitochondria of *Parl*[-/-] spermatocytes appear consistently swollen with few thin irregular cristae and loss of normal matrix density (*Figure 2C*). Quantitative analysis of the mitochondrial ultrastructure in primary spermatocytes showed a dramatic increase of degenerating mitochondria in *Parl*[-/-] compared to WT spermatocytes (92% of analyzed mitochondria in *Parl*[-/-] were abnormal vs. 1.9% in WT; n = 3 for each phenotype; p=0.0002 by two-sided Fisher's exact test). Importantly, abnormal mitochondrial morphology was the earliest ultrastructural change detected in PARL-deficient spermatocytes localized in the abluminal compartment, while adluminal germ cells exhibited additional abnormalities affecting other cell compartments, including the endoplasmic reticulum, Golgi apparatus, and nuclear envelope. Chromatin clumping and nuclear fragmentation were also evident (*Figure 2B* and *Figure 2—figure supplement 1A*).

In contrast, other cell types within the seminiferous tubules and surrounding interstitium, such as spermatogonia, Leydig, and Sertoli cells, displayed normal ultrastructural features with preserved mitochondrial morphology (*Figure 2—figure supplement 1B*). Altogether, these data indicate the presence of early mitochondrial ultrastructural abnormalities culminating in extensive degeneration and death of arrested PARL-deficient spermatocytes, without morphological evidence of spermatogonia, Leydig cells, or Sertoli cells involvement.

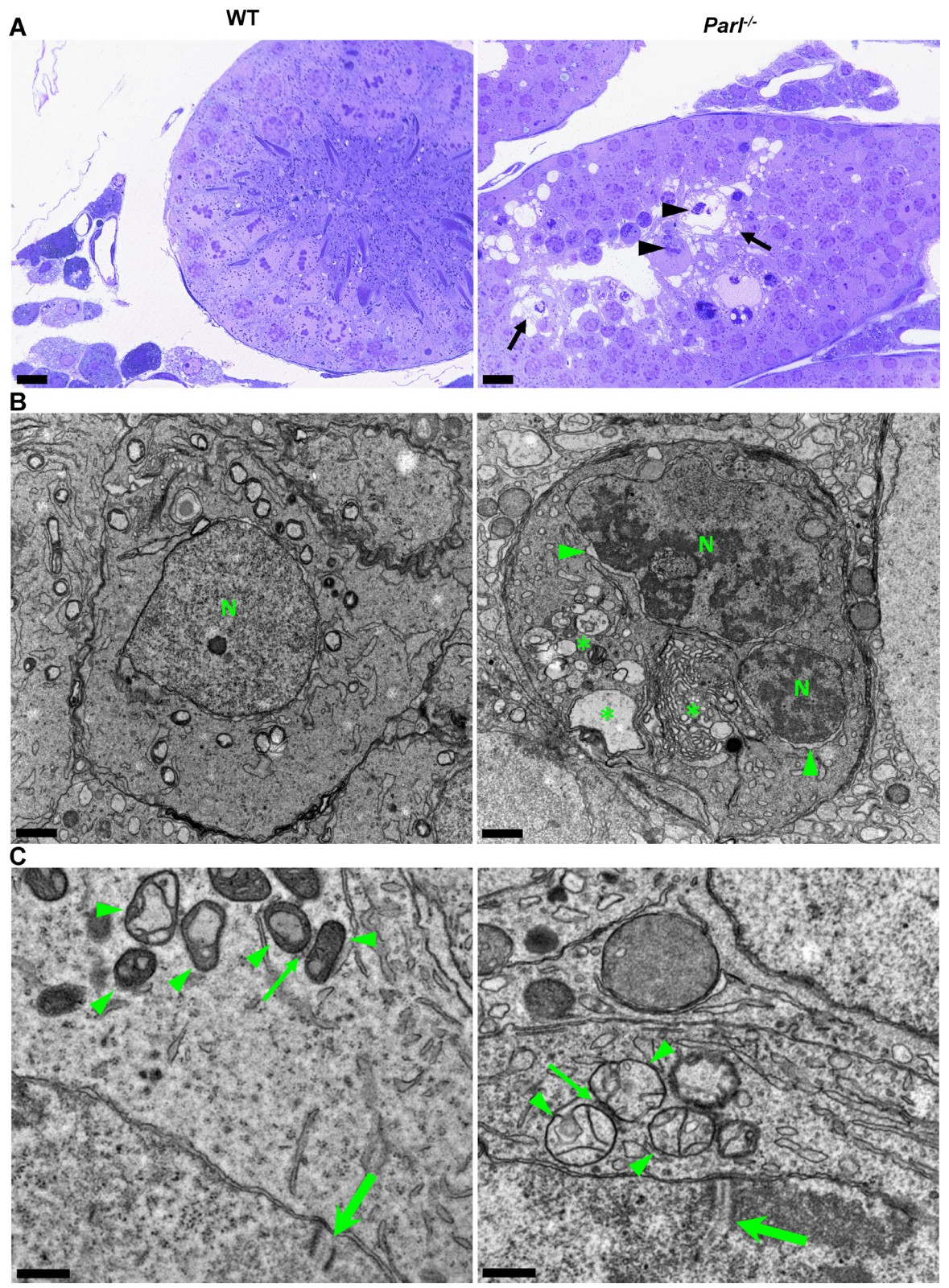

**Figure 2.** Impaired spermatogenesis in *Parl*⁻/⁻ testis is associated with early mitochondrial morphological abnormalities and progressive degeneration of arrested spermatocytes. (**A**) Toluidine blue-stained semithin sections of testis from 5-week-old WT and *Parl*⁻/⁻ mice. Seminiferous tubules from *Parl*⁻/⁻ mice show extensive degenerative changes in arrested spermatocytes including tortuous membrane infoldings, cytoplasmic vacuolation (arrows), irregular chromatin clumping, nuclear fragmentation (arrowheads), and absence of mature germ cells such as adluminal spermatids and spermatozoa

*Figure 2 continued on next page*

*Figure 2 continued*

(n = 3 for each genotype). A WT seminiferous tubule with normal germ cell maturation is shown for comparison (left panel). Scale bars, 20 μm. (**B**) Electron microscopy examination shows multifocal cisternae distention, disruption of the endoplasmic reticulum and Golgi apparatus, and abundant accumulation of damaged membranous material and organelles (asterisks) in *Parl*⁻/⁻ spermatocytes. The nuclear envelope is diffusely distended (arrowheads) outlining a convoluted fragmented nucleus (N) with dense irregular clumps of chromatin. A WT spermatocyte at the end of pachytene is shown for comparison (left panel). Scale bars, 1 μm. (**C**) Electron microscopy analysis shows that mitochondria in *Parl*⁻/⁻ primary spermatocytes are swollen with few thin irregular cristae and loss of normal matrix density (right panel, arrowheads) compared to WT (left panel, arrowheads). The thin arrows indicate the intermitochondrial cement (nuage) typically associated with mitochondria in primary spermatocytes. The large arrows indicate fully assembled synaptonemal complexes, structures that are only detectable during the zygotene and pachytene stages of meiotic prophase I (n = 3 for each genotype). Scale bars, 0.5 μm.

The online version of this article includes the following figure supplement(s) for figure 2:

**Figure supplement 1.** Ultrastructural abnormalities of mitochondria and other cell compartments are restricted to arrested spermatocytes and absent in other testis cell types.

## Impaired spermatogenesis in PARL-deficient testis is not driven by misprocessing of PARL substrates PINK1, PGAM5, and TTC19

Next, we asked to what extent the severe spermatogenesis defect induced by PARL deficiency can be attributed to the misprocessing and altered maturation of PARL's substrates. To answer this question, we first tested the testicular expression of established PARL substrates. *Parl*⁻/⁻ testis mitochondria exhibit remarkable accumulation of uncleaved PINK1 and PGAM5, as well as almost total lack of the mature form of TTC19 (*Figure 3A*). These findings were consistent with previous observations in the brain (*Spinazzi et al., 2019*) and cultured cells (*Saita et al., 2017*). Since other PARL substrates, such as DIABLO, STARD7, and CLPB, displayed only subtle misprocessing or expression changes, possibly due to compensatory proteolytic cleavage by alternative proteases, we focused our investigation on PINK1, PGAM5, and TTC19. We aimed to determine whether the genetic modulation of these substrates could either modify or reproduce the testicular phenotype observed in *Parl*⁻/⁻ mice. In particular, we assessed whether accumulation of uncleaved PINK1 and PGAM5, alone or in combination, or depletion of the cleaved form of PINK1, PGAM5, or TTC19 were the molecular mechanisms underlying the abnormalities documented in *Parl*⁻/⁻ testis. PINK1 and PGAM5 are known to play essential roles in maintaining mitochondrial integrity and homeostasis and have been linked to both Parkinson's disease and defects of spermatogenesis (*Valente et al., 2004*; *Lu et al., 2014*; *Agarwal et al., 2020*; *Deng et al., 2008*). Similarly, TTC19 is a mitochondrial protein crucial for the catalytic activity of complex III, and pathogenic variants of TTC19 are associated with mitochondrial diseases in humans, including Leigh syndrome (*Atwal, 2014*).

To test this hypothesis, we analyzed testes from a series of genetically engineered mutant mouse lines, including single-gene knockouts such as *Pink1*⁻/⁻, *Pgam5*⁻/⁻, and *Ttc19*⁻/⁻, as well as multiple gene knockouts including both *Parl* and *Pink1* (*Parl*⁻/⁻/*Pink1*⁻/⁻); *Parl* and *Pgam5* (*Parl*⁻/⁻/*Pgam5*⁻/⁻); *Pink1* and *Pgam5* (*Pink1*⁻/⁻/*Pgam5*⁻/⁻); and *Parl*, *Pink1*, and *Pgam5* combined (*Parl*⁻/⁻/*Pink1*⁻/⁻/*Pgam5*⁻/⁻). Remarkably, the severe testis phenotype resulting from PARL deficiency remained unaltered upon additional deletion of *Pink1* or *Pgam5* either individually or in combination (*Figure 3B* and *Figure 3—figure supplement 1*). In contrast, the single or combined knockouts of *Pink1*, *Pgam5*, and *Ttc19* resulted in normal fertility and testis morphology, showing orderly and complete spermatogenesis (*Figure 3B* and *Figure 3—figure supplement 1*). In conclusion, these observations indicate that impaired spermatogenesis in PARL-deficient mice is not driven by altered proteolytic maturation of the substrates PINK1, PGAM5, and TTC19 despite their severely affected proteolytic processing, indicating that other pathogenetic mechanisms are responsible for the testis phenotype.

## PARL-deficient testis mitochondria exhibit severe respiratory chain defects

Spermatogenesis involves crucial metabolic adaptations, with mitochondrial function playing a critical role throughout germ cell maturation (*Varuzhanyan and Chan, 2020*). Given the interconnection between mitochondrial morphology and function, we investigated the impact of the structural abnormalities identified in the mitochondria of *Parl*⁻/⁻ spermatocytes by conducting a comprehensive

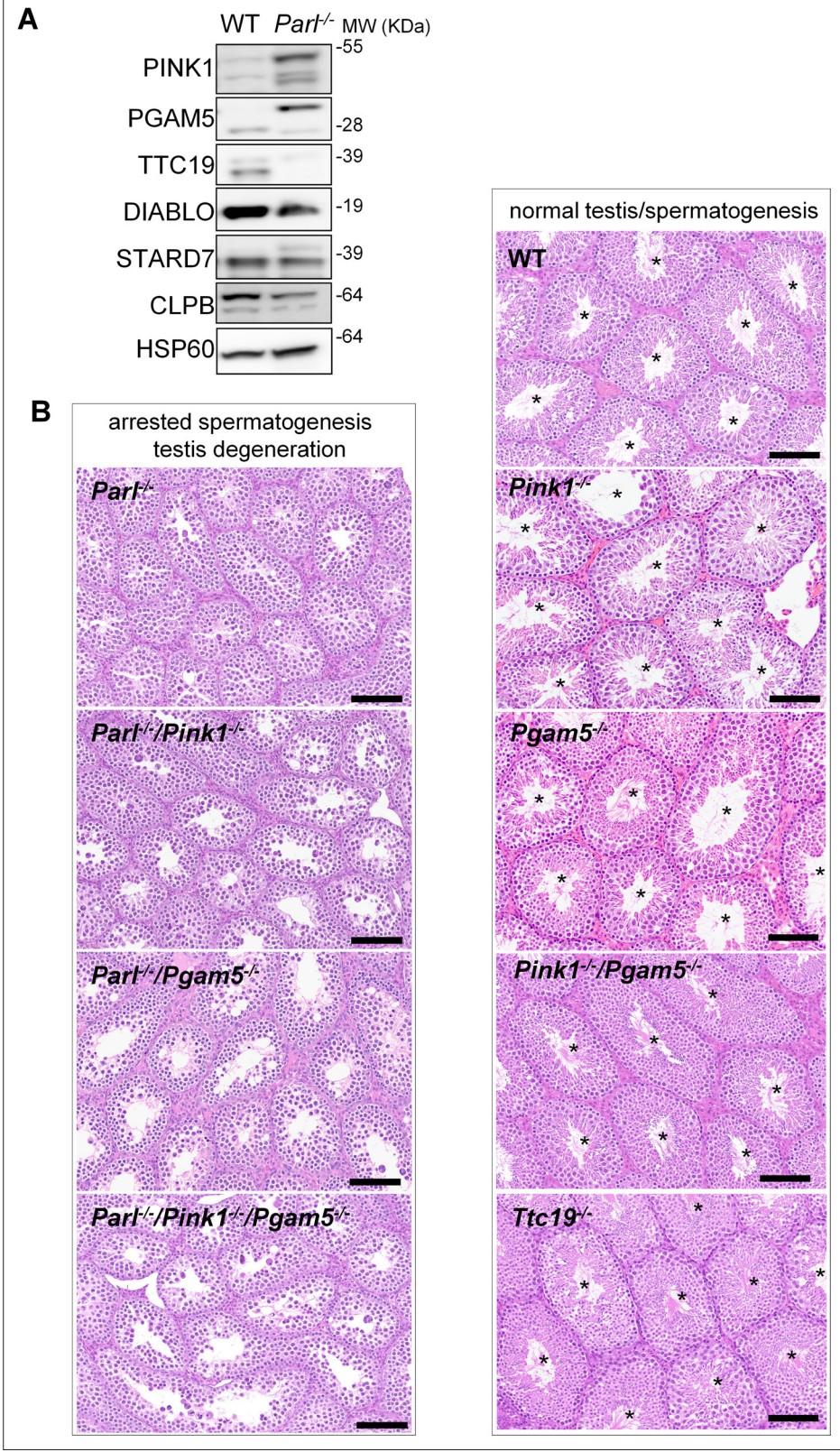

**Figure 3.** Mice with genetic manipulation of the PARL substrates PINK1, PGAM5, and TTC19 do not reproduce or modify *Parl⁻/⁻* testis phenotype. (**A**) Immunoblots of testis mitochondria from 6-week-old WT and *Parl⁻/⁻* mice with antibodies for the established PARL substrates PINK1, PGAM5, TTC19, DIABLO, STARD7, and CLPB. Severe accumulation of unprocessed PINK1 and PGAM5, as well as severe decrease in the mature processed form of

*Figure 3 continued on next page*

*Figure 3 continued*

TTC19 are evident in *Parl⁻/⁻* testis. HSP60 is the loading control. (**B**) Histology of testes from 7-week-old mice of the indicated genotypes (HE stain, n = 3 for each genotype). *Parl⁻/⁻/Pink1⁻/⁻*, *Parl⁻/⁻/Pgam5⁻/⁻*, and *Parl⁻/⁻/Pink1⁻/⁻/Pgam5⁻/⁻* show complete lack of sperm production and no modification of the testicular phenotype compared to *Parl⁻/⁻* mice. *Ttc19⁻/⁻*, *Pink1⁻/⁻*, *Pgam5⁻/⁻*, and *Pink1⁻/⁻/Pgam5⁻/⁻* mice have no evident testis pathology and show normal sperm production (mature spermatozoa are indicated by asterisks), and are fertile. Scale bar, 145 μm.

The online version of this article includes the following source data and figure supplement(s) for figure 3:

**Source data 1.** Original images for *Figure 3A*.

**Figure supplement 1.** Similar to what has been described in *Parl⁻/⁻* mice, immunohistochemistry for AIF-1 confirms spermatogenesis arrest with complete absence of spermatids in *Parl⁻/⁻/Pink1⁻/⁻*, *Parl⁻/⁻/Pgam5⁻/⁻*, and *Parl⁻/⁻/Pink1⁻/⁻/Pgam5⁻/⁻* mice.

functional analysis. Because of the previously reported role of PARL in mitochondrial biogenesis (*Civitarese et al., 2010*), we wondered whether mitochondrial mass is reduced in *Parl⁻/⁻* testis. Expression of the outer mitochondrial membrane protein TOMM20 and of the inner membrane ATP synthase beta subunit (ATPB) were similar between WT and *Parl⁻/⁻* testis, suggesting unaltered mitochondrial mass (*Figure 4A*, *Figure 5B*, and *Figure 6B*). Similarly, mitochondrial DNA abundance, often used as an indicator of mitochondrial mass, was not significantly different between the two groups (*Figure 4B*). Additionally, the expression of TFAM, a protein binding mitochondrial DNA in nucleoids (*Ekstrand et al., 2004*), did not show any significant difference between WT and *Parl⁻/⁻* SCP-1-positive spermatocytes (*Figure 4—figure supplement 1*). Next, we examined whether mitochondrial respiratory chain complexes were appropriately assembled in *Parl⁻/⁻* testis mitochondria. Blue native gel electrophoresis revealed severe assembly alterations in multiple respiratory chain complexes, including complex I, complex III, complex IV, and to a lesser extent complex V, as well as the supercomplex (*Pérez-Pérez et al., 2016*; *Figure 4C*). Since respiratory chain complexes' supramolecular assembly is required for optimizing the efficiency of mitochondrial oxidative phosphorylation (OXPHOS), we then examined if PARL deficiency ultimately resulted in impaired mitochondrial respiration in testis mitochondria. To answer this question, we measured oxygen consumption by means of high-resolution respirometry in testis mitochondria supplied with substrates and specific inhibitors for complex I (CI), complex II (CII), and complex IV (CIV) as illustrated in *Figure 4D*. Basal mitochondrial respiration in presence of complex I substrates but no ADP (CI LEAK) was significantly increased in *Parl⁻/⁻* testis compared to WT, suggesting pathological short-circuit of protons across the inner mitochondrial membrane. Conversely, both phosphorylating respiration, whether driven by complex I only (CI OXPHOS) or by both complex I and II together (CI + II OXPHOS), and maximal uncoupled respiration, whether driven by complex II (CII ET) or by both complex I and II (CI + II ET) were severely diminished in *Parl⁻/⁻* testis mitochondria. Respiration driven by CIV was also decreased. These results localize the severe respiration defect at the level of electron transfer capacity (*Figure 4E*). However, the defects were not attributed to cytochrome *c* loss due to outer mitochondrial membrane permeabilization (*Figure 4E*; CIV+cytc graph). To gain cell-type insights into the observed electron transport defect, cytochrome *c*-oxidase activity staining was performed on frozen tissue sections. The enzyme function was significantly decreased in PARL-deficient seminiferous tubules but not in Leydig cells, highlighting the specific distribution of the defect (*Figure 4F*). The expression of the subunit 4 of cytochrome *c*-oxidase, COX4, was indeed severely decreased in *Parl⁻/⁻* SCP-1-positive spermatocytes, confirming the defect in this cell type (*Figure 5A*; p=0.0027). This defect was again unrelated to changes in mitochondrial mass since TOMM20 expression was unmodified by PARL deficiency in SCP-1 spermatocytes (*Figure 5B*). Interestingly, e dramatic overexpression of the glucose intracellular transporter GLUT1 was observed in *Parl⁻/⁻* spermatocytes suggesting increased glucose utilization as an adaptive response to disrupted OXPHOS (*Figure 5C*). In conclusion, PARL is crucial for maintaining the integrity of the mitochondrial electron transport chain. Its deficiency leads to severe respiratory chain defects and metabolic remodeling in arrested primary spermatocytes.

## PARL deficiency causes impaired testicular CoQ biogenesis and redox

CoQ is a lipid essential for cellular functions, serving both as an electron carrier in the mitochondrial respiratory chain and as a lipophilic antioxidant, preventing lipid peroxidation (*Gueguen et al., 2021*). In mammalian mitochondria, CoQ is involved in multiple converging pathways for its reduction,

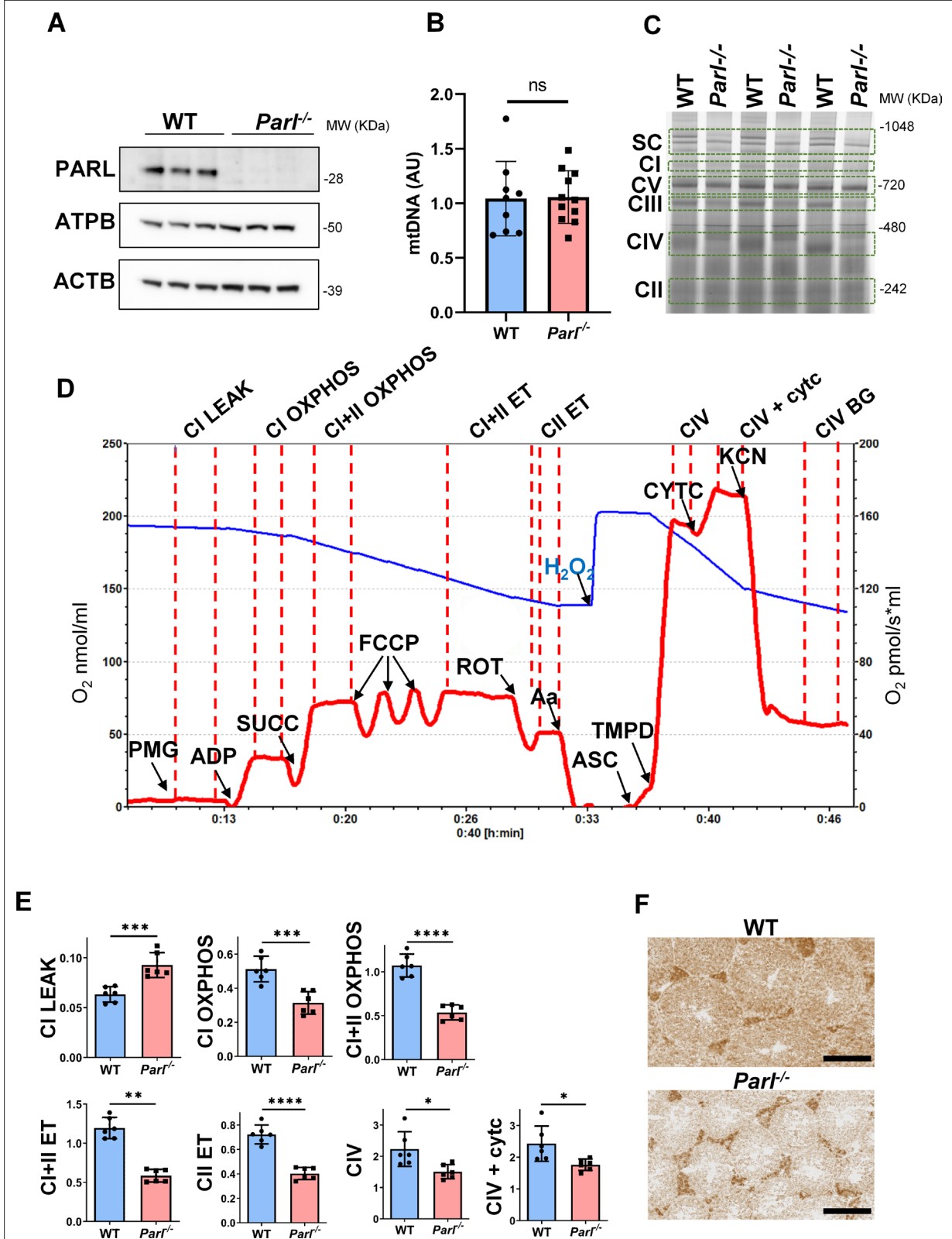

**Figure 4.** Severe mitochondrial electron transfer defects in *Parl*⁻ᐟ⁻ testis mitochondria. (**A**) Immunoblots of testis lysates from 5-week-old WT and *Parl*⁻ᐟ⁻ mice with antibodies for PARL, ATPB, TOMM20, and ACTB (n = 3 for each genotype). ACTB is the loading control. (**B**) Quantification of mitochondrial DNA normalized to nuclear DNA in testis from 5-week-old WT and *Parl*⁻ᐟ⁻ mice (n = 10 for each genotype). MtDNA was quantified by measuring the ratio (mtDNA/nDNA) between a target mitochondrial gene (*Cox1*) and a reference nuclear gene (*B2m*) using quantitative real-time PCR as detailed in the

*Figure 4 continued on next page*

*Figure 4 continued*

'Methods' section. No significant difference is found between WT and *Parl*$^{-/-}$ testis (p=0.9146). (**C**) Blue native gel electrophoresis of testis mitochondria from 6-week-old WT and *Parl*$^{-/-}$ mice (n = 3 for each genotype). Mitochondrial complexes and supercomplex constituted by macromolecular assembly of complex I (CI), complex III (CIII) dimer, and complex IV (CIV) are visualized after staining with Instant Blue and marked by dotted lines. Assembly defects are evident for CI, CIII, CIV, and the supercomplex. (**D**) Representative trace illustrating the protocol for high-resolution respirometry in testis mitochondria. The blue trace indicates the O$_2$ concentration (nmol/ml), and the red trace indicates its time derivative (pmol of O$_2$ consumed/s*ml). Testis mitochondria (150 µg) were loaded in Miro6 buffer. Substrates are as follows: CI (PMG, pyruvate + malate + glutamate), CII (Succ, succinate), and CIV (ASC/TMPD, ascorbate + TMPD). The uncoupler is CCCP. The specific mitochondrial inhibitors are rotenone (ROT) for CI, antimycin a (Aa) for CIII, and cyanide (KCN) for CIV. Respiratory states are indicated between red dashed lines. CI LEAK, CI-driven leak respiration, in presence of CI substrates but no adenylates; CI OXPHOS, CI-driven phosphorylating respiration; CI+II OXPHOS, phosphorylating respiration driven by combined activation of CI and II; CI+II ET, electron transfer capacity driven by combined CI and II; CII ET, ET driven by CII; CIV, CIV-driven respiration; CIV+cytc: CIV-driven respiration after addition of exogenous cytochrome *c* to evaluate the integrity of the outer mitochondrial membranes; CIV BG: chemical background of CIV-driven respiration. H$_2$O$_2$ in the presence of catalase is used to reoxygenate the chamber. (**E**) Quantification of the respiratory states of testis mitochondria from 6-week-old WT and *Parl*$^{-/-}$ mice (n = 6 for each genotype) as from the protocol described in (**D**) and in the 'Methods' section. Bar graphs indicate average ± SD. Statistical significance calculated by two-sided Student's *t*-test: *p<0.05, **p<0.01,***p<0.001, and ****p<0.0001. (**F**) Cytochrome *c* oxidase histochemistry in frozen testis sections from 6-week-old WT and *Parl*$^{-/-}$ mice (n = 3 for each genotype).

The online version of this article includes the following source data and figure supplement(s) for figure 4:

**Source data 1.** Original images for ***Figure 4A***.

**Figure supplement 1.** Unaltered TFAM expression in *Parl*$^{-/-}$ spermatocytes.

including complex I, complex II, dehydro-orotate dehydrogenase, sulfide-quinone oxidoreductase, and electron transfer dehydrogenase, while complex III is responsible for its oxidation. CoQ plays a critical role in promoting testicular functions including the maturation of male germ cells by safeguarding against oxidative damage (***Lin et al., 2021***; ***Mancini and Balercia, 2011***).

In previous studies, we showed that brain mitochondria from PARL-deficient mice have decreased CoQ concentration linked to impaired expression of the ubiquinone biosynthesis protein COQ4 homolog, mitochondrial COQ4 (***Spinazzi et al., 2019***), a protein required for the biosynthesis of CoQ (***Wang and Hekimi, 2019***). Additionally, we observed an increase in the reduced-to-oxidized CoQ ratio (CoQ red/ox) in neurons due to TTC19 deficiency, leading to complex III dysfunction (***Spinazzi et al., 2019***). Similarly, we found significantly decreased CoQ levels in *Parl*$^{-/-}$ testis, accompanied by a dramatic increase in the CoQ red/ox (***Figure 6A***). This elevation in CoQ reduction can be attributed to impaired CoQH$_2$ oxidation, resulting from compromised complex III activity caused by TTC19 depletion (***Figure 3A***) and complex III assembly defects (***Figure 4C***). Notably, we also noticed a substantial decrease in COQ4 levels in *Parl*$^{-/-}$ testis, as seen in the brain (***Spinazzi et al., 2019***). Western blotting (***Figure 6B***) and immunohistochemistry (***Figure 6C***) revealed a diffuse decrease in COQ4 expression in various cell types, including germ cells, Leydig cells, and Sertoli cells. The deficit was particularly pronounced in *Parl*$^{-/-}$ arrested spermatocytes, even those with no or minimal degenerative changes, suggesting that the CoQ biosynthesis defect occurred upstream of the degenerative process. Quantitative immunofluorescence of COQ4 expression confirms severe deficiency of this protein in *Parl*$^{-/-}$ SCP-1-positive spermatocytes compared to WT littermates (***Figure 6—figure supplement 1***). Collectively, our findings indicate that PARL plays a crucial role in maintaining CoQ biosynthesis and redox state.

## PARL deficiency leads to ferroptosis in arrested spermatocytes

To understand the specific mechanism responsible for the severe germ cell degeneration and death observed in PARL-deficient mice, we first considered apoptosis due to the characteristic ultrastructural features observed in arrested spermatocytes (i.e., chromatin clumping and nuclear fragmentation) and previous links of PARL to antiapoptotic properties *in vitro* (***Cipolat et al., 2006***). However, levels of caspase-3 activation in the seminiferous tubules of *Parl*$^{-/-}$ mice were comparable to WT, suggesting that apoptosis was not significantly involved in this phenotype (***Figure 7—figure supplement 1***). Given the identification of decreased CoQ concentration and severe ultrastructural abnormalities involving mitochondria and other membranous cell compartments, we speculated about the possible role of ferroptosis. Ferroptosis is a programmed cell death modality characterized by lipid peroxidation of cell membranes (***Stockwell et al., 2017***; ***Santoro, 2020***). Previous studies in cultured cells have shown the importance of CoQ producing mevalonate pathway (***Shimada et al., 2016***) and CoQ reducing pathways driven by FSP1 (***Bersuker et al., 2019***; ***Doll et al., 2019***), DHODH (***Mao et al., 2021***), and GCH1 (***Kraft et al., 2020***) in ferroptosis. To test this hypothesis, we examined the expression of

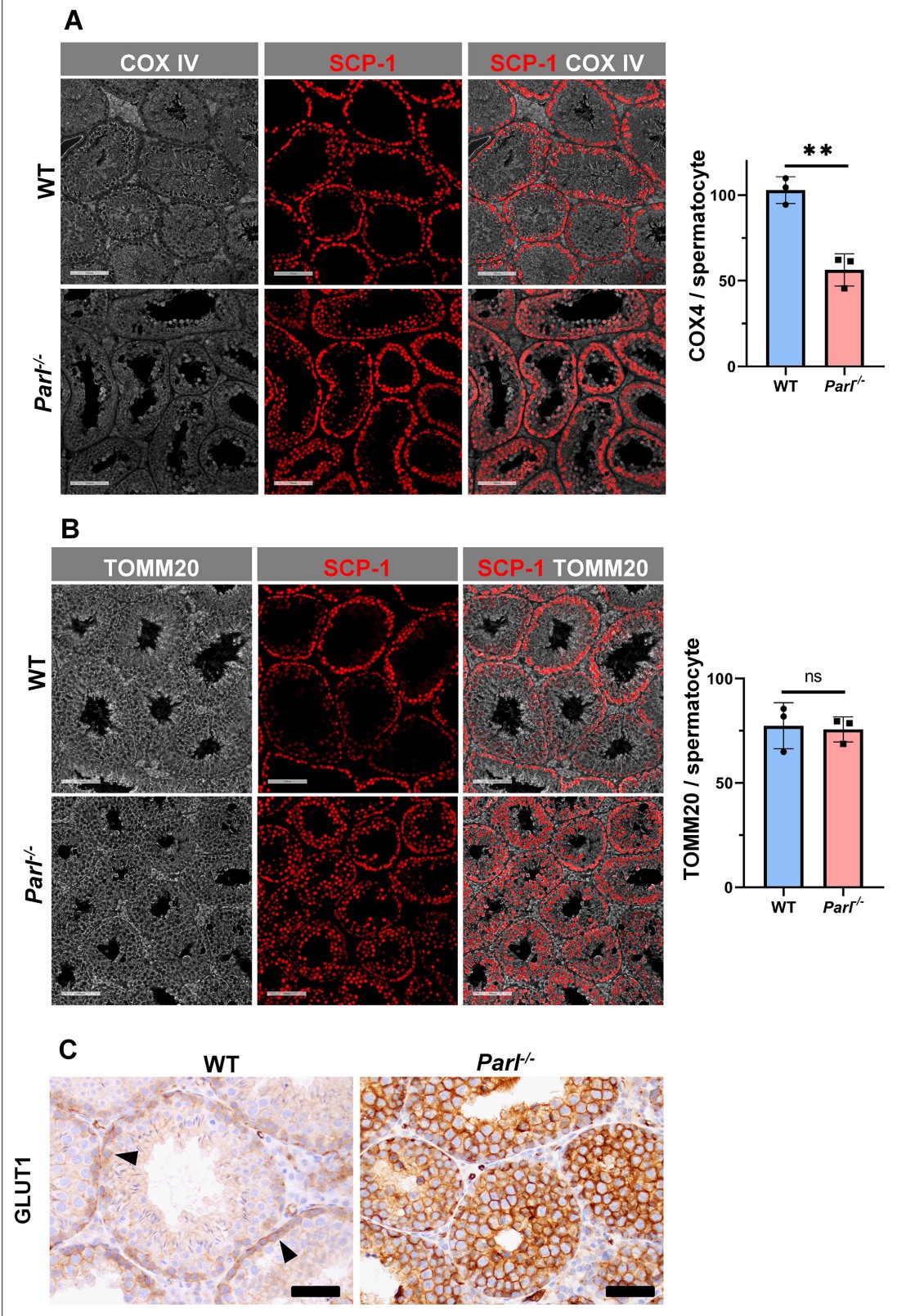

**Figure 5.** Severe loss of COX4 associated with increased expression of glucose intracellular transporter in *Parl*<sup>-/-</sup> spermatocytes. (**A**) Quantitative immunofluorescence shows decreased expression of COX4 in SCP-1-positive spermatocytes from 5-week-old *Parl*<sup>-/-</sup> mice compared to WT littermates (n = 3 for each genotype, 500–1000 SCP-1-positive spermatocytes for each mouse, two-sided Student's *t*-test: p=0.0027). Scale bars, 100 µm. Bar graphs indicate average ± SD. (**B**) Normalized quantification of TOMM20 immunofluorescence in SCP-1-positive primary spermatocytes does not reveal

*Figure 5 continued on next page*

*Figure 5 continued*

significant differences in mitochondrial mass in the two different genotypes (n = 3 mice for each genotype, 500–1000 SCP-1-positive spermatocytes considered for each mouse; p=0.821). Scale bars, 100 μm. Bar graphs indicate average ± SD. Statistical significance calculated by two-sided Student's *t*-test. (**C**) GLUT1 immunohistochemistry of testis from 5-week-old mice shows prominent overexpression of GLUT1 in arrested *Parl*[-/-] spermatocytes, and low levels in WT (arrowheads) (n = 3 for each genotype). Scale bars, 50 μm.

GPX4, a crucial antioxidant peroxidase that prevents ferroptosis by reducing phospholipid hydroperoxide in cell membranes using reduced glutathione as substrate (*Chen et al., 2021*; *Seibt et al., 2019*). Immunoblot analysis revealed a nearly complete absence of GPX4 expression in *Parl*[-/-] testis (*Figure 7A*). Immunohistochemistry and immunofluorescence provided cell-type-specific insights, showing a dramatic decrease in GPX4 expression in *Parl*[-/-] arrested spermatocytes (*Figure 7C*, top panels; *Figure 8A*, p=0.0013) but not in Leydig (*Figure 7C*, top panels, black arrowheads) or Sertoli cells (*Figure 8—figure supplement 1*; p=0.5313). The impact of PARL deficiency on GPX4 expression was not observed in other organs, indicating a specific effect on spermatocytes (*Figure 7—figure supplement 2A*). To rule out a possible effect of PARL proteolytic activity on GPX4 expression, we checked GPX4 expression in mouse embryonic fibroblasts with and without PARL expression, and knockouts rescued with proteolytically active or inactive PARL. The results do not show evidence of proteolytic misprocessing and do not indicate GPX4 as a direct substrate of PARL (*Figure 7—figure supplement 2B*). Further investigations demonstrated increased lipid peroxidation, as evidenced by significantly higher levels of 4-hydroxynonenal (HNE) adducts, the end-products of lipid peroxidation that defines ferroptosis, in *Parl*[-/-] testis (*Figure 7B*, middle panel), but not in brain (*Figure 7—figure supplement 2C*). The accumulation of HNE adducts was particularly prominent in adluminal and exfoliated spermatocytes during the late stages of degeneration (*Figure 7C*, middle panels). We confirmed these data by quantitative immunofluorescence showing a dramatic increase in HNE signal in SCP-1 positive *Parl*[-/-] spermatocytes (*Figure 8B*; p=0.0002), which is consistent with the specific loss of GPX4 expression in these cells.

Additional established biomarkers of ferroptosis, including cellular tumor antigen p53, a master regulator of both canonical and non-canonical ferroptosis pathways (*Jiang et al., 2015*; *Liu and Gu, 2022*), and transferrin receptor protein 1 (TfR1), which promotes the cellular uptake of iron via receptor-mediated endocytosis (*Feng et al., 2020*), were also investigated. Excessive intracellular iron can contribute to ferroptosis by triggering lipid peroxidation through Fenton's reaction. In normal postpubertal mice, expression of p53 levels in testis is very low (*Beumer et al., 1998*) while TfR1 is very high in spermatogonia and gradually decreases during germ cell maturation (*Leichtmann-Bardoogo et al., 2012*; *Gao et al., 2021*; *Figure 7C*, bottom panel). In contrast, PARL-deficient testis showed prominent nuclear expression of p53 in adluminal degenerating spermatocytes (*Figure 7—figure supplement 3*), while TfR1 exhibited persistent overexpression in arrested spermatocytes (*Figure 7C*, bottom panel), suggesting abnormally high iron uptake. We confirmed these data by quantitative immunofluorescence showing increased TfR1 expression in *Parl*[-/-] SCP-1-positive spermatocytes compared to WT littermates (*Figure 8—figure supplement 2*; p=0.0229). These findings collectively indicate that ferroptosis is a cell-type-specific effect of PARL deficiency and the mechanism underlying the demise of *Parl*[-/-] arrested spermatocytes.

## Discussion

This study sheds light on the critical role of PARL in spermatogenesis and germ cell survival by maintaining the mitochondrial respiratory chain, CoQ biogenesis, and regulating ferroptosis. The reported testicular phenotype represents the earliest manifestation of PARL deficiency. Interestingly, similar spermatogenic defects and neurodegeneration have been reported in *Drosophila* mutants lacking the mitochondrial rhomboid orthologue Rhomboid-7, suggesting that the physiological roles of the mitochondrial rhomboid in these tissues are conserved across different phyla in the animal kingdom (*McQuibban et al., 2006*; *Spinazzi et al., 2019*). Impaired spermatogenesis in the *Parl*[-/-] mouse is characterized by a complete maturation arrest before the completion of the first meiotic division, leading to the induction of ferroptosis in primary spermatocytes. This meiotic failure seems to be related to severe morphological abnormalities of mitochondria and respiratory chain defects. This finding reinforces the crucial role of mitochondrial fitness in supporting germ cell differentiation during spermatogenesis,

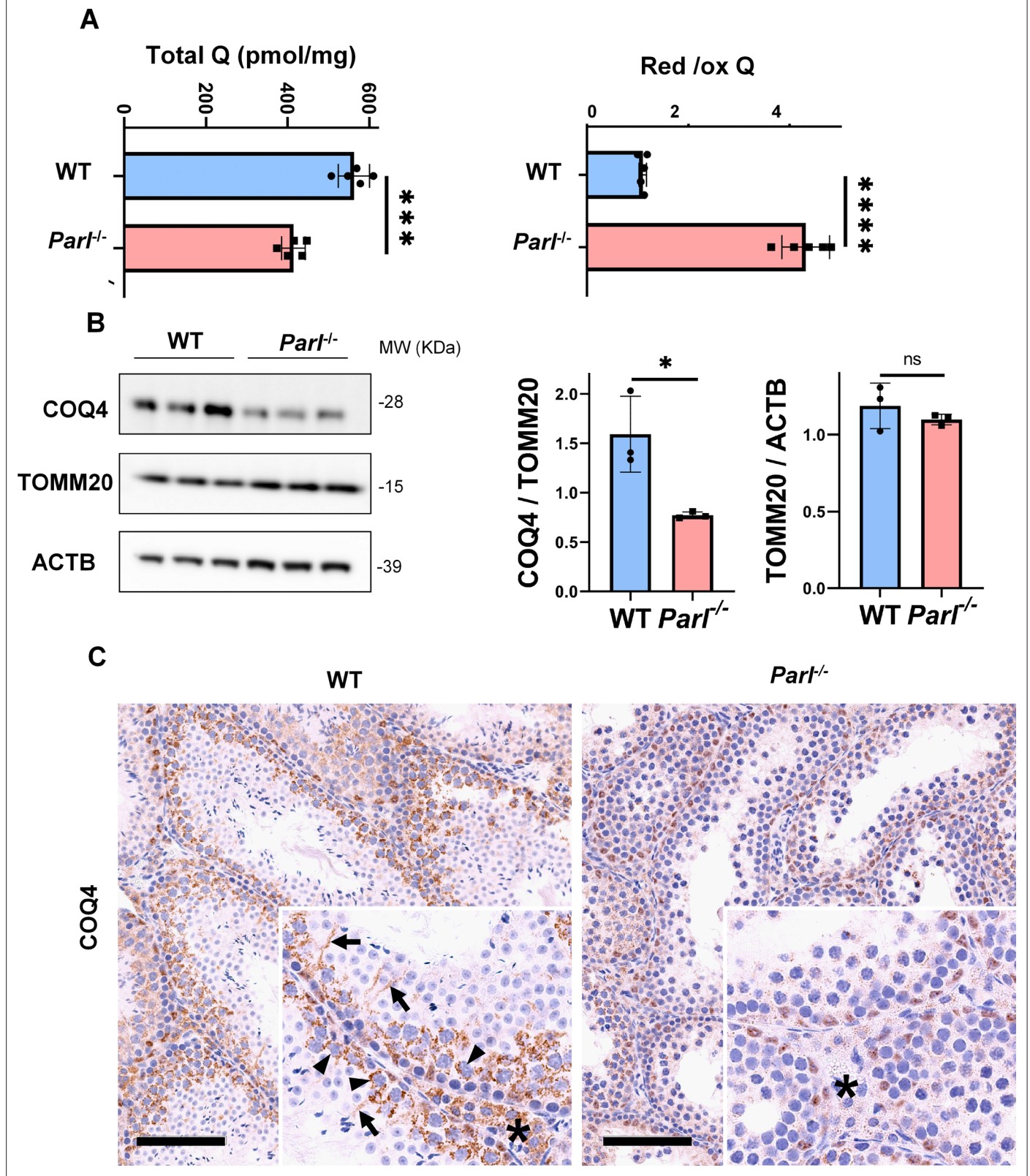

**Figure 6.** Severe alteration in coenzyme Q (CoQ) biosynthesis and redox state in *Parl⁻/⁻* testis. (**A**) Concentration (left) and CoQ red/ox ratio (right) of total CoQ ($Q_9 + Q_{10}$) measured by HPLC in the testes of 5-week-old WT and *Parl⁻/⁻* mice (n = 5 for each genotype). Total CoQ levels are severely decreased in *Parl⁻/⁻* testis compared to WT littermates (p=0.0001 calculated by two-sided Student's *t*-test). Moreover, the redox status is altered with drastic elevation in the reduced/oxidized CoQ ratio (p<0,0001 calculated by two-sided Student's *t*-test). (**B**) Immunoblot analysis of total testis lysates

*Figure 6 continued on next page*

*Figure 6 continued*

from 5-week-old WT and *Parl*⁻/⁻ mice with antibodies for COQ4, TOMM20, and ACTB (n = 3 for each genotype). ACTB is the total lysate loading control. TOMM20 is the mitochondrial content control. Quantification of COQ4/TOMM20 confirms a significant decrease in *Parl*⁻/⁻ testis compared to WT littermates (n = 3; p=0,0212 calculated by two-sided Student's *t*-test.) but unchanged TOMM20/ACTB (n = 3; p=0,368 calculated by two-sided Student's *t*-test), indicating that the observed decrease in COQ expression is not explained by decreased mitochondrial mass. Bar graphs indicate average ± SD. (**C**) Immunohistochemistry for COQ4 shows severely decreased levels of testicular COQ4 expression in 5-week-old *Parl*⁻/⁻ mice compared to WT controls (n = 3 for each genotype). The deficit is particularly prominent in *Parl*⁻/⁻ arrested spermatocytes, almost devoid of COQ4 expression, compared to the high constitutive levels of COQ4 expression in WT spermatocytes (inset, stage II tubule, arrowheads). Decreased COQ4 expression is also evident in *Parl*⁻/⁻ Leydig cells compared to WT mice (insets, asterisk). In addition, COQ4-positive Sertoli cell projections observed in WT mice (inset, stage II tubule, arrows) are not evident in the seminiferous tubules of *Parl*⁻/⁻ mice. Scale bar, 100 µm.

The online version of this article includes the following source data and figure supplement(s) for figure 6:

**Source data 1.** Original images for *Figure 6B*.

**Figure supplement 1.** Severe loss of COQ4 in *Parl*⁻/⁻ spermatocytes.

as previously observed in other mouse models with mitochondrial impairment including defective mitochondrial DNA (*Trifunovic et al., 2004*; *Nakada et al., 2006*), adenylates transport (*Brower et al., 2009*), cardiolipin biosynthesis (*Cadalbert et al., 2015*), mitochondrial dynamics (*Varuzhanyan et al., 2019*; *Varuzhanyan et al., 2021*), and mitochondrial proteolysis (*Gispert et al., 2013*; *Lu et al., 2008*). In our model, respiratory chain defects involve the assembly and function of multiple complexes, as well as the biosynthesis of the electron carrier CoQ. These results corroborate earlier observations in *Parl*⁻/⁻ brain (*Spinazzi et al., 2019*) and recently published studies confirming impaired CoQ biogenesis in *PARL*⁻/⁻ cell culture models (*Deshwal et al., 2023*). Collectively, these data underscore a crucial but previously underestimated role of PARL in maintaining the respiratory chain, CoQ biosynthesis, and mitochondrial structure (*Spinazzi et al., 2019*).

The reason for the pronounced respiratory chain defects and mitochondrial abnormalities in spermatocytes compared to other cell types is not entirely clear. However, we speculate that these differences may arise from cell-type-specific metabolic requirements. Normal spermatogenesis requires a significant metabolic remodeling, with a shift from glycolysis to oxidative phosphorylation to support the energy demand for completing the first meiotic division (*Wang et al., 2022*). In the absence of PARL, primary spermatocytes seem unable to implement oxidative phosphorylation due to their defective respiratory chain, leading to meiotic arrest, despite compensating with increased intracellular glucose uptake . These findings suggest that this phenotype is mainly driven by a germ cell-autonomous defect. Further investigations using germ-cell-specific *Parl* conditional knockout mice may help elucidate the contribution of somatic cells to this phenotype.

PARL deficiency in spermatocytes leads to maturation arrest and progressive degeneration of germ cells, culminating in the activation of ferroptosis, a specific type of regulated necrosis (*Seibt et al., 2019*). While PARL's essential role in cell survival has been established (*Spinazzi and De Strooper, 2016*), its relationship with apoptosis remains contradictory in cellular models (*Saita et al., 2017*; *Cipolat et al., 2006*). Recent findings indicate that PARL deficiency induces necrosis rather than apoptosis in the brain (*Spinazzi et al., 2019*). Although we cannot rule out the contribution of accidental necrosis, since no specific markers are actually available for this cell death modality, this study highlights the specific induction of ferroptosis as the primary mechanism leading to the demise of PARL-deficient spermatocytes.

Ferroptosis represents a specific type of regulated cell death, characterized by uncontrolled iron-dependent lipid peroxidation of cell membranes (*Chen et al., 2021*). The presence of ferroptosis in PARL-deficient spermatocytes is evidenced by the dramatic accumulation of HNE, an electrophilic aldehyde generated by lipid peroxidation, and impaired expression of the ferroptosis suppressor GPX4. While ferroptosis has been documented in germ cells from *Caenorhabditis elegans* (*Perez et al., 2020*), it has not been extensively studied in mammalian spermatogenesis. In this context, our study provides evidence *in vivo* for the implication of ferroptosis during impaired spermatogenesis in a mammalian model. Ferroptosis can be experimentally induced *in vitro* by chemical or genetic inhibition of GPX4, or depletion of its substrate glutathione (*Zheng and Conrad, 2020*; *Stockwell et al., 2017*). Although much of what is known today about ferroptosis comes from *in vitro* experiments or studies in organisms with genetic inactivation of GPX4, its pathophysiological implication in diseases is rapidly emerging (*Stockwell, 2022*). GPX4 exists in three distinct isoforms originating from

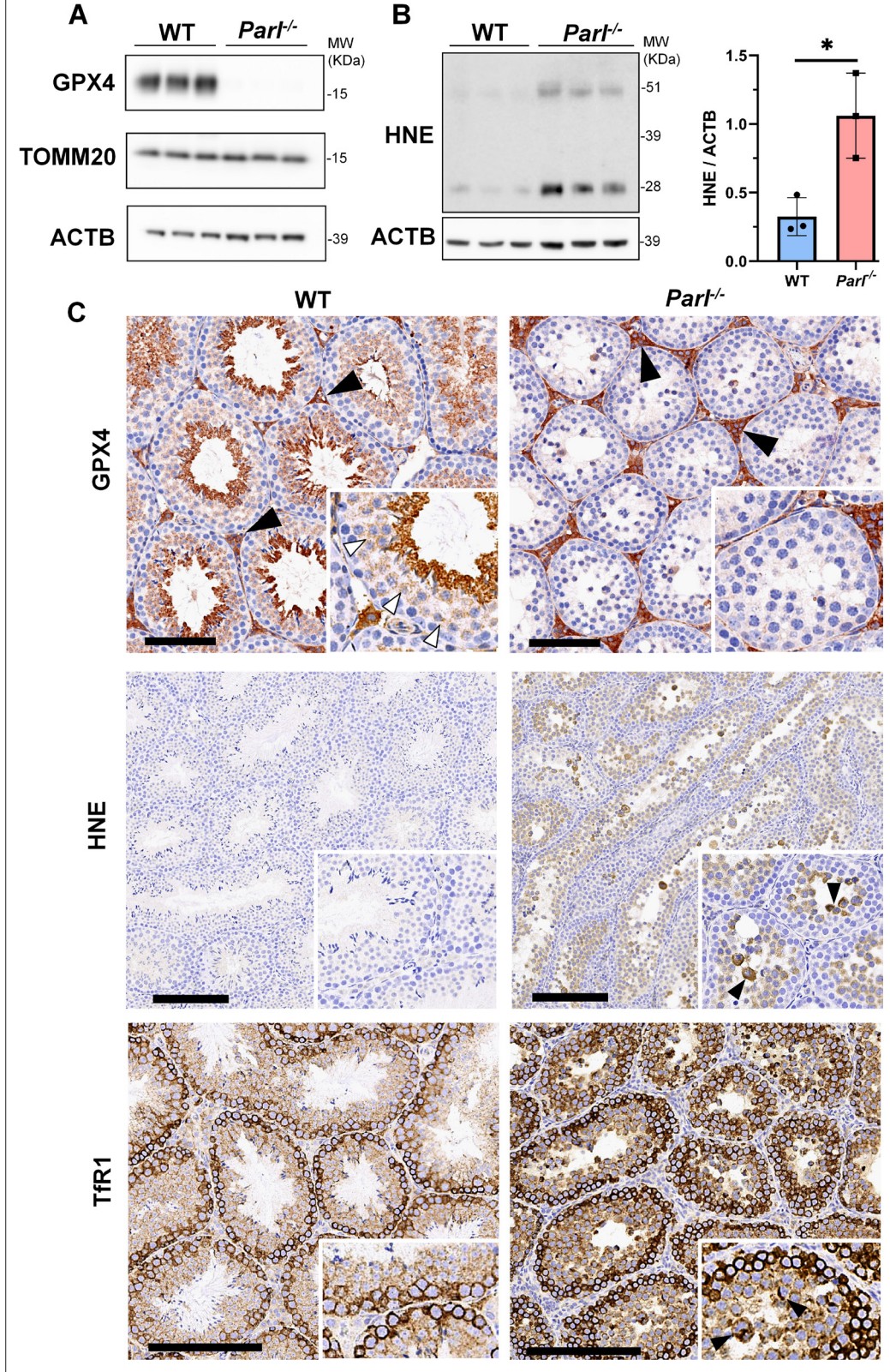

**Figure 7.** Massive ferroptosis activation in *Parl*<sup>-/-</sup> arrested spermatocytes. (**A**) Immunoblot of total testis lysates obtained from 5-week-old WT and *Parl*<sup>-/-</sup> mice using antibodies for GPX4, TOMM20, and ACTB (n = 3 for each genotype). ACTB is the loading control. GPX4 expression is barely detectable in *Parl*<sup>-/-</sup> testis. (**B**) Immunoblot analysis of total testis lysates from 7-week-old WT and *Parl*<sup>-/-</sup> mice using anti-HNE and anti-ACTB antibodies

*Figure 7 continued on next page*

*Figure 7 continued*

(n = 3 for each genotype). ACTB is the loading control. Quantification of the HNE/ACTB ratio is shown on the right as a graph indicating average ± SD (n = 3 for each genotype). The statistically significant HNE/ACTB ratio increase in *Parl*[-/-] mice has been calculated by two-sided Student's *t*-test (p=0.0199). (**C**) Immunohistochemistry for GPX4, HNE, and TfR1 in testis from 6-week-old WT and *Parl*[-/-] mice (n = 3 for each genotype). GPX4 expression is barely detectable in *Parl*[-/-] arrested spermatocytes compared to WT littermates (inset, stage X tubule, white arrowheads), while it is unaffected in interstitial Leydig cells (black arrowheads) (top panels, scale bar, 100 um). HNE immunohistochemistry shows gradual intensification of lipid peroxidation during spermatocyte degeneration culminating in adluminal/exfoliated spermatocytes (inset, arrowheads) (middle panels; scale bar, 200 μm). Similarly, TfR1 expression is abnormally increased in degenerating *Parl*[-/-] spermatocytes (bottom panels; scale bars, 200 μm; inset, arrowheads).

The online version of this article includes the following source data and figure supplement(s) for figure 7:

**Source data 1.** Original images for *Figure 7A*.

**Source data 2.** Original images for *Figure 7B*.

**Figure supplement 1.** Unremarkable levels of apoptosis activation in degenerated *Parl*[-/-] testis.

**Figure supplement 2.** Lack of effect of PARL proteolytic activity on GPX4 expression *in vitro* and testis-specific induction of ferroptosis in PARL-deficient mice.

**Figure supplement 2—source data 1.** Original images for *Figure 7—figure supplement 2A*.

**Figure supplement 2—source data 2.** Original images for *Figure 7—figure supplement 2B*.

**Figure supplement 2—source data 3.** Original images for *Figure 7—figure supplement 2C*.

**Figure supplement 3.** Increased activation of p53 in PARL-deficient spermatocytes undergoing ferroptosis.

**Figure supplement 3—source data 1.** Original images for *Figure 7—figure supplement 3A*.

different transcription initiation sites: a full-length mitochondrial form, a shorter cytosolic form, and a nuclear isoform (*Maiorino et al., 2003*). GPX4 expression is highest in testis, where the mitochondrial isoform is mainly expressed (*Godeas et al., 1997*). Germline deletion of *Gpx4* in mice results in embryonic lethality (*Yant et al., 2003*), while tissue-specific deletions lead to premature death (*Seiler et al., 2008*; *Tan et al., 2021*; *Friedmann Angeli et al., 2014*; *Carlson et al., 2016*; *Wortmann et al., 2013*). Notably, spermatocyte-specific *Gpx4* deletion in mice causes severe testicular atrophy, reduced spermatogenesis, germ cell death, and infertility (*Imai et al., 2009*), highlighting its importance in male reproductive biology. Reduced GPX4 activity is also observed in the sperm of infertile patients, emphasizing its role in human spermatogenesis (*Imai et al., 2001*; *Foresta et al., 2002*; *Hao et al., 2023*).

In addition to GPX4, other defense mechanisms against ferroptosis have been described, including CoQ, which provides powerful protection from lipid peroxidation in cell membranes (*Gueguen et al., 2021*; *Bersuker et al., 2019*; *Doll et al., 2019*; *Mao et al., 2021*; *Tan et al., 2021*). Although the contribution of mitochondria to ferroptosis is still being debated (*Zheng and Conrad, 2020*), cumulating evidence indicates that mitochondria are implicated in this process (*Gao et al., 2019*). CoQ is in fact most abundant in mitochondria, where its biosynthesis takes place, and from which CoQ is then distributed to other cell membranes including the plasma membrane, Golgi apparatus, and endoplasmic reticulum (*Gueguen et al., 2021*; *Stefely and Pagliarini, 2017*). Moreover, in cancer cells treated with GPX4 inhibitors to induce ferroptosis, dihydroorotate dehydrogenase DHODH, a mitochondrial inner membrane enzyme involved in pyrimidine biosynthesis, inhibits ferroptosis by reducing CoQ (*Mao et al., 2021*), suggesting that mitochondrial CoQ reduction inhibits ferroptosis. Although the lack of GPX4 is *per se* sufficient to induce ferroptosis in *Parl*[-/-] spermatocytes, the process appears exacerbated by the concomitant CoQ deficiency. The functional interaction between PARL, GPX4, and CoQ in the determination of ferroptosis is consistent with a recent study, published during the revision of our manuscript, reporting increased susceptibility of *PARL*[-/-] cultured cells to ferroptosis induction by GPX4 inhibitors (*Deshwal et al., 2023*). The underlying mechanism involved defective CoQ biosynthesis and intracellular distribution outside mitochondria mediated by the PARL substrate STARD7 (*Deshwal et al., 2023*). Altogether, these converging results demonstrate the implication of PARL in the regulation of ferroptosis in specific conditions, both *in vitro* and *in vivo*. Interestingly, GPX4 is not a PARL substrate, hence the mechanism beyond GPX4 loss in this cell type remains currently unclear. One possibility is that GPX4 deficiency may result from protein degradation linked

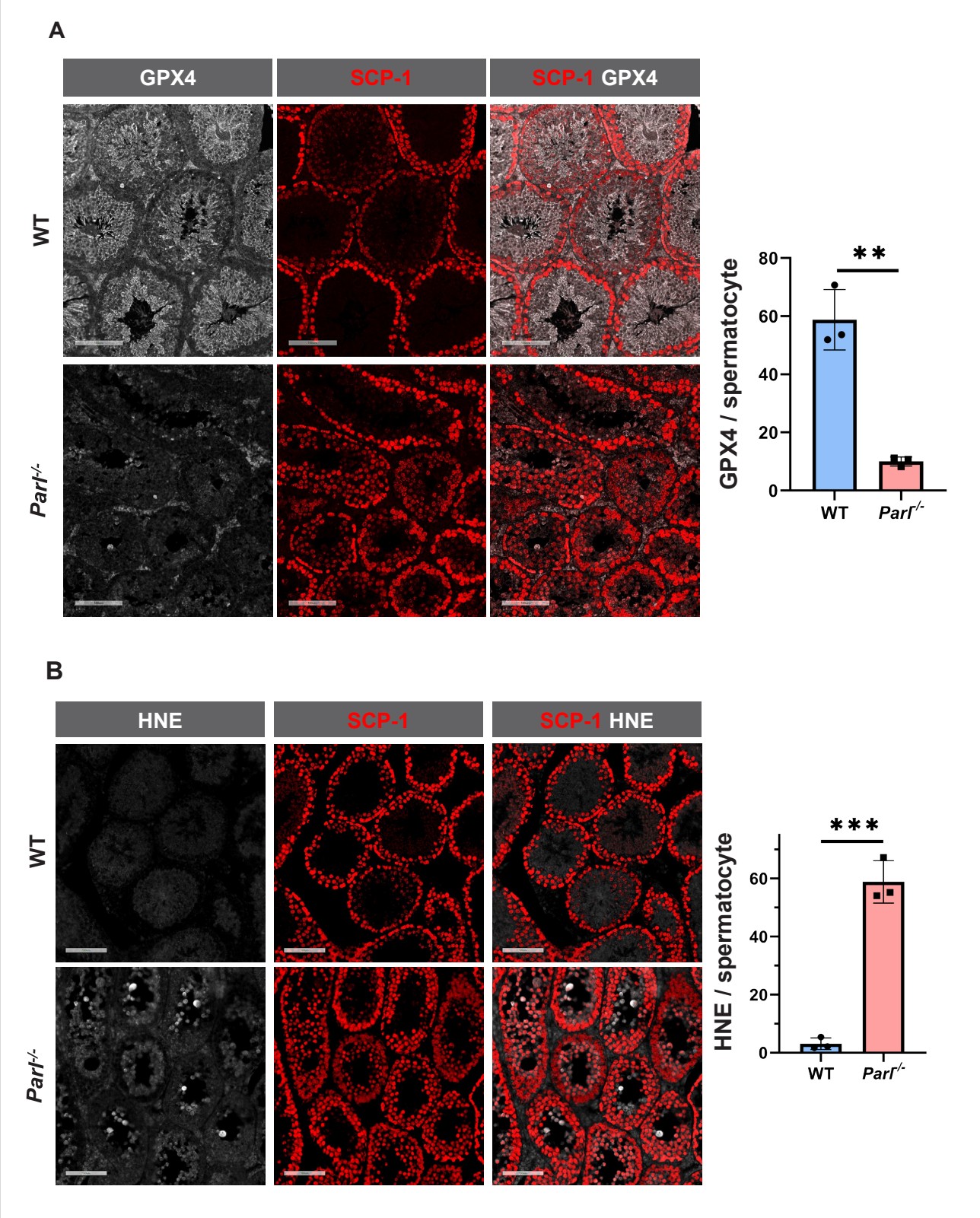

**Figure 8.** Hallmarks of ferroptosis in *Parl*-/- primary spermatocytes. (**A**) Quantitative immunofluorescence shows severely reduced GPX4 expression in SCP-1-positive primary spermatocytes from 5-week-old *Parl*-/- mice compared to WT littermates (n = 3 mice for each genotype, 500–1000 SCP-1-positive spermatocytes considered for each mouse, p=0.0013). (**B**) Quantitative immunofluorescence shows increased HNE accumulation in *Parl*-/- SCP-1-positive

*Figure 8 continued on next page*

*Figure 8 continued*

spermatocytes compared to WT littermates (n = 3 mice for each genotype, 500–1000 SCP-1-positive spermatocytes considered for each mouse, p=0.0002). Bar graphs indicate average ± SD. Statistical significance calculated by two-sided Student's *t*-test. Scale bars, 100 μm.

The online version of this article includes the following figure supplement(s) for figure 8:

**Figure supplement 1.** Unchanged GPX4 expression in *Parl⁻/⁻* Sertoli cells.

**Figure supplement 2.** Increased transferrin receptor expression in *Parl⁻/⁻* spermatocytes undergoing ferroptosis.

to chaperon-mediated autophagy, as reported in cells treated with the ferroptosis inducer erastin (***Wu et al., 2019***), but we cannot rule out a spermatocyte-specific effect on *Gpx4* gene expression either. Interestingly, some interdependence between GPX4 and CoQ is suggested by overlapping inhibitory effects of the ferroptosis inducer FIN56 (***Shimada et al., 2016***) on both CoQ and GPX4 and by the influence of mevalonate pathway on the isopentenylation of selenocysteine-tRNA (***Moosmann and Behl, 2004***) needed for efficient GPX4 expression. Moreover, GPX4 deficit has been previously found in the brain of CoQ-deficient *Coq9^{R239X}* mice (***Luna-Sánchez et al., 2017***), suggesting that GPX4 loss and ferroptosis may be an overlooked mechanism of CoQ deficiency deserving further investigations. The reason why only spermatocytes undergo ferroptosis in absence of PARL is likely related to the specific loss of GPX4 expression. The distinct vulnerability of spermatocytes might also be influenced by the high poly-unsaturated fatty acid content in these cells (***Oresti et al., 2010***). This peculiar feature could render spermatocytes exceptionally susceptible to lipid peroxidation in the context of the observed CoQ deficiency. This observation provides an important example of how specific phenotypes of mitochondrial diseases can be caused by unexpected cell-type-specific pathophysiological mechanisms downstream of mitochondrial dysfunction. Similar observations can provide some

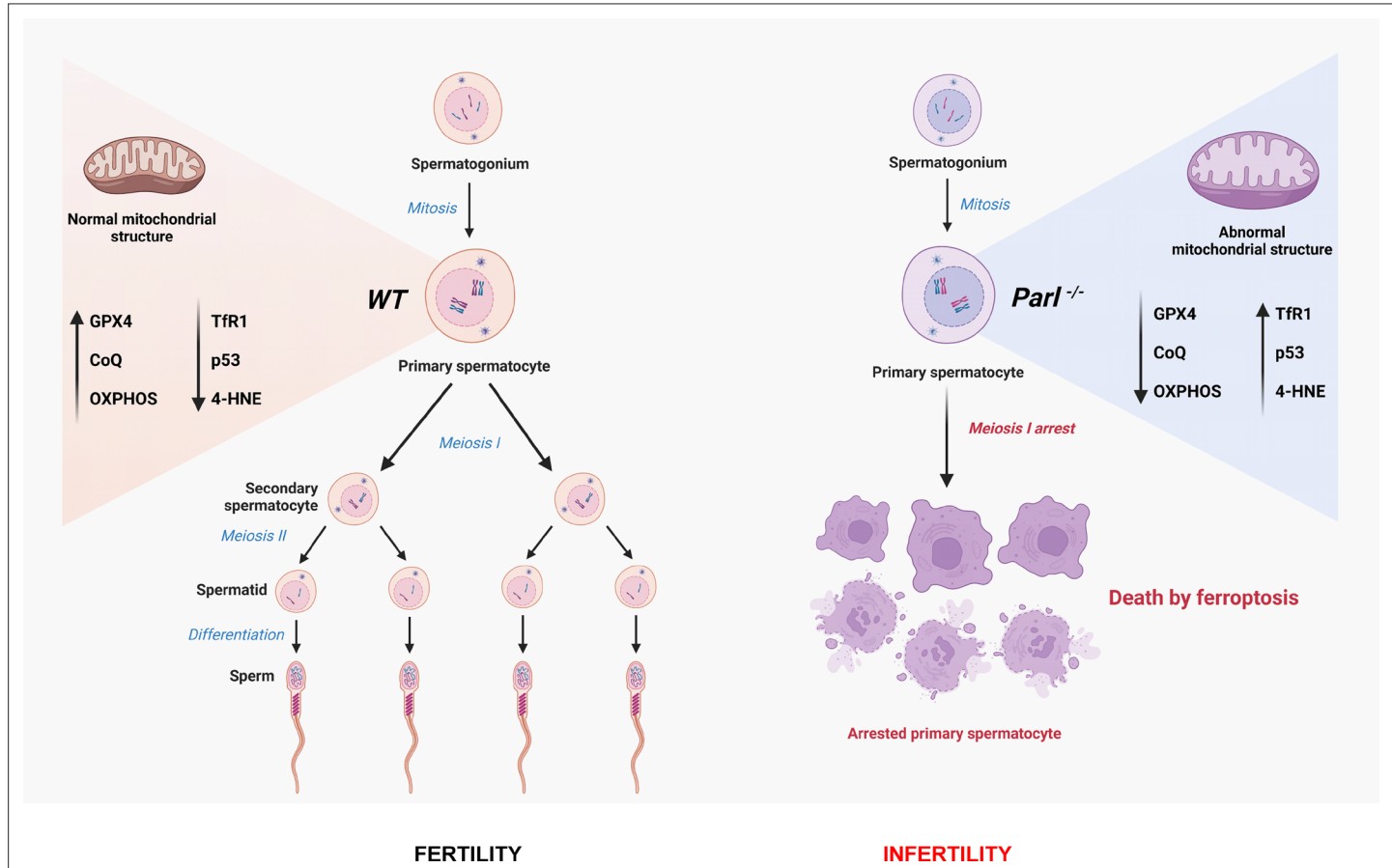

**Figure 9.** Cartoon illustrating the identified mechanisms underlying the spermatogenesis defect of *Parl⁻/⁻* mice and the induction of spermatocyte ferroptosis.

explanations for our very limited understanding of the tissue-specific clinical manifestations of mitochondrial diseases.

Knowledge of the physiological relevance of ferroptosis in mitochondrial diseases is limited. Compensatory activation of ferroptosis-inhibitory pathways has been recently reported in some conditions of mitochondrial deficiencies. In hearts from mice with different types of mitochondrial dysfunction, such as mitochondrial genome expression defects (*Kühl et al., 2017*) or cytochrome *c* oxidase deficiency (*Ahola et al., 2022*), GPX4 expression increases. Ahola and collaborators have elegantly shown that upregulation of GPX4, sustained by increased glutathione metabolism via the trans-sulphuration pathway and improved selenium incorporation in GPX4, provides a crucial homeostatic response to prevent ferroptosis in heart tissue of $Cox10^{-/-}$ mice (*Ahola et al., 2022*). Impairing GPX4 upregulation induced by OXPHOS deficiency through the inhibition of the integrated stress response, by knocking out either the mitochondrial protease OMA1 or its substrate DELE1, aggravates the cardiomyopathy of $Cox10^{-/-}$ mice by decreasing GPX4 to WT levels, thus inducing ferroptosis (*Ahola et al., 2022*). Moreover, direct ablation of GPX4 in cultured cells affected by defective OXPHOS induced by a variety of mitochondrial respiratory chain inhibitors is lethal (*To et al., 2019*). These data clearly demonstrate that prevention of ferroptosis by means of GPX4 upregulation is a required physiological mechanism to prevent cell death in conditions of defective OXPHOS. Our study provides *in vivo* evidence for this mechanism by showing the opposite situation: ferroptosis is spontaneously initiated in PARL-deficient spermatocytes unable to tune up ferroptosis inhibitory pathways in response to OXPHOS deficiency.

In conclusion, this work establishes PARL's crucial role in spermatogenesis and prevention of germ cell ferroptosis by maintaining the integrity of mitochondrial structure, electron transport chain, CoQ biosynthesis, and GPX4 expression in spermatocytes (*Figure 9*). The discovery of ferroptosis as a consequence of primary mitochondrial defects advances our understanding of the pathophysiology of mitochondrial diseases and male infertility, offering potential targets for future therapeutic interventions.

# Methods

## Key resources table

| Reagent type (species) or resource | Designation | Source or reference | Identifiers | Additional information |
|---|---|---|---|---|
| Genetic reagent (*Mus musculus*) | B6.129P2(Cg)-*Parl*$^{tm1.1Bdes}$/Ieg | PMID:16839884 | RRID:IMSR_EM:02075 | |
| Genetic reagent (*M. musculus*) | B6.129P2-*Parl*$^{tm1Bdes}$/Ieg | PMID:16839884 | RRID:IMSR_EM:02076 | |
| Genetic reagent (*M. musculus*) | Tg(Nes-cre)1Kln; *Parl*$^{tm1Bdes}$/*Parl*$^{tm1Bdes}$ | PMID:10471508 | RRID:MGI:6280694 | |
| Genetic reagent (*M. musculus*) | *Parl*$^{tm1.1Bdes}$/*Parl*$^{tm1.1Bdes}$; *Pgam5*$^{tm1d(EUCOMM)Wtsi}$/*Pgam5*$^{tm1d(EUCOMM)Wtsi}$ | PMID:30578322 | RRID:MGI:6280688 | |
| Genetic reagent (*M. musculus*) | *Parl*$^{tm1.1Bdes}$/*Parl*$^{tm1.1Bdes}$; *Pink1*$^{tm1.1Wrst}$/*Pink1*$^{tm1.1Wrst}$ | PMID:20049710 | RRID:MGI:6280687 | |
| Genetic reagent (*M. musculus*) | C57BL/6J-*Ttc19*$^{em1Bds}$ | PMID:30578322 | RRID:MGI:6280684 | |
| Cell line (*M. musculus*) | WT, *Parl*-/-, *Parl*-/-+*Parl*$^{WT}$, *Parl*-/-+*Parl*$^{S275A}$ | PMID:30578322 | RRID:MGI:2159769 | |
| Antibody | Anti-SCP-1 (rabbit monoclonal) | Abcam | Cat#: ab175191 | IHC 1:200, IF 1:500 |
| Antibody | Anti-AIF1 (rabbit polyclonal) | Wako | Cat#: 019-19741; RRID:AB_839504 | IHC 1:200 |
| Antibody | Anti-COQ4 (rabbit polyclonal) | ProteinTech | Cat#: 16654-1AP; RRID:AB_2878296 | IF 1:800, IHC 1:200, WB 1:1000 |
| Antibody | Anti-GPX4 (rabbit polyclonal) | Sigma | Cat#: HPA047224; RRID:AB_2679990 | IHC 1:100, IF 1:500 |

*Continued on next page*

*Continued*

| Reagent type (species) or resource | Designation | Source or reference | Identifiers | Additional information |
|---|---|---|---|---|
| Antibody | Anti-HNE (rabbit polyclonal) | Alpha Diagnostic International | Cat#: HNE11-S; RRID:AB_2629282 | IF 1:10,000, IHC 1:3000 |
| Antibody | Anti-TFR1 (rabbit monoclonal) | Abcam | Cat#: ab214039; RRID:AB_2904534 | IF 1:3000, IHC 1:1000 |
| Antibody | Anti-p53 (rabbit polyclonal) | Leica/Novocastra | Cat#: NCL-L-p53-CM5p; RRID:AB_2895247 | IHC 1:300 |
| Antibody | Anti-p53 (mouse monoclonal) | Cell Signaling Technology | (1C12) Mouse mAb #2524; RRID:AB_331743 | WB 1:1000 |
| Antibody | Anti-Wilm's tumor 1 (rabbit monoclonal) | Abcam | Cat#: ab89901; RRID:AB_2043201 | IF 1:1500 |
| Antibody | Anti-cKit (rabbit polyclonal) | Agilent/DAKO | Cat#: A4502; RRID:AB_2335702 | IHC 1:50 |
| Antibody | Anti-γH2AX (rabbit monoclonal) | Cell Signaling Technology | Cat#: 2577; RRID:AB_2118010 | IHC 1:1000 |
| Antibody | Anti-COX4 (rabbit polyclonal) | ProteinTech | Cat#:11242–1-AP; RRID:AB_2085278 | IF1:3000 |
| Antibody | Anti-GLUT1 (rabbit monoclonal) | Cell Signaling Technology | Cat#:73015 | IHC 1:600 |
| Antibody | Anti-TOMM20 (rabbit polyclonal) | ProteinTech | Cat#: 73015; RRID:AB_2207530 | IF 1:4000 |
| Antibody | Anti-TFAM (rabbit polyclonal) | Abcam | Cat#: ab307302 | IF 1:3000 |
| Antibody | Anti-PARL (rabbit polyclonal) | PMID:16839884 | Cat#: N/A | WB 1/1000 |
| Antibody | Anti-actin (mouse monoclonal) | Sigma | Cat#: A5441; RRID:AB_476744 | WB 1:200,000 |
| Antibody | Anti-HSP60 (mouse monoclonal) | BD Biosciences | Cat#: 611562; RRID:AB_399008 | WB 1:50,000 |
| Antibody | Anti-ATP5B (mouse monoclonal) | Abcam | Cat#: ab14730; RRID:AB_301438 | WB 1:50,000 |
| Antibody | Anti-TOMM20 (rabbit polyclonal) | Santa Cruz | Cat#: sc-11415; RRID:AB_2207533 | WB 1:5000 |
| Antibody | Anti-PINK1 (rabbit polyclonal) | Cayman | Cat#: 10006283; RRID:AB_10098326 | WB 1:500 |
| Antibody | Anti-PGAM5 (rabbit polyclonal) | Sigma | Cat#: HPA036979; RRID:AB_10960559 | WB 1:250 |
| Antibody | Anti-TTC19 (rabbit polyclonal) | Sigma | Cat#: HPA052380; RRID:AB_2681806 | WB 1:2000 |
| Antibody | Anti-CLPB (rabbit polyclonal) | ProteinTech | Cat#: 15743-1-AP; RRID:AB_2847900 | WB 1:1000 |
| Antibody | Anti-STARD7 (rabbit polyclonal) | ProteinTech | Cat#: 15689-1-AP; RRID:AB_2197820 | WB 1:2000 |
| Antibody | Anti-DIABLO (rabbit polyclonal) | Cell Signaling Technology | Cat#: 15108; RRID:AB_2798711 | WB 1:1000 |
| Antibody | Anti-GPX4 (mouse monoclonal) | R&D Systems | Cat#: MAB5457; RRID:AB_2232542 | WB 1:1000 |
| Antibody | Anti-GPX4 (mouse monoclonal) | Santa Cruz | Cat#: sc-166570; RRID:AB_2112427 | WB 1:1000 |

*Continued on next page*

Continued

| Reagent type (species) or resource | Designation | Source or reference | Identifiers | Additional information |
|---|---|---|---|---|
| Antibody | Anti-HNE (mouse monoclonal) | R&D Systems | Cat#: 198960; RRID:AB_664165 | WB 1:500 |
| Antibody | Anti-citrate synthase (mouse monoclonal) | Abcam | Cat#: ab96600; RRID:AB_10678258 | WB 1:1000 |
| Software, algorithm | GraphPad Prism software | GraphPad Prism (https://graphpad.com) | RRID:SCR_015807 | |
| Software | ImageJ software | ImageJ (http://imagej.nih.gov/ij/) | RRID:SCR_003070 | |

## Animals and husbandry

Mice with full knockout germline deletion of *Parl* (*Parl*$^{-/-}$) (MGI:3693645), *Pgam5* (*Pgam5*$^{-/-}$) (MGI:5882561), *Pink1* (*Pink1*$^{-/-}$) (MGI:5436308), *Ttc19* (*Ttc19*$^{-/-}$) (MGI:6276545), and conditional *Parl* ablation under the Nestin promoter (*Parl* $^{L/L}$::*Nes*$^{Cre}$) (MGI:3526574, MGI:2176173) have been generated as previously described (*Spinazzi et al., 2019*; *Cipolat et al., 2006*). All mutant mouse lines were maintained on a C57BL/6J background. Mice were kept in a SPF facility and multiply housed in filter top polycarbonated cages enriched with wood-wool and shavings as bedding. Standard rodent diet and acidified tap water were provided *ad libitum*. Animal rooms were maintained at 22°C ± 2°C with a 45 and 70% relative humidity range, 50 air changes per hour, and 12-hr light/dark cycles. Mice were included in a health-monitoring program developed in accordance with the guidelines of the Federation of European Laboratory Animal Science Associations (FELASA). All experiments were approved by the Ethical Committee on Animal Experimenting of the University of Leuven (IACUC protocol #072/2015) and the French Ministry (DUO-OGM 5769 3/2019).

## Pathological and immunohistochemical examination

Testes harvested from postpubertal mutant mice and WT matched controls were immersion-fixed in 10% neutral buffered formalin for 24–48 hr at room temperature (RT). Samples were then routinely processed for paraffin embedding, sectioned at 5 μm, and stained with hematoxylin and eosin (HE) for histopathological assessment. For immunohistochemistry (IHC), 5-μm-thick paraffin sections were mounted on ProbeOn slides (Thermo Fisher Scientific #15-188-51). Chromogenic immunohistochemistry (IHC) and multiplex immunofluorescence (IF) were performed as described elsewhere (*Tarrant et al., 2021*) using a Leica BOND RXm automated platform combined with the Bond Polymer Refine Detection kit (Leica #DS9800) for IHC or the OPAL Automation Multiplex IHC Detection Kit (Akoya Biosciences NEL830001KT) implemented onto a Leica BOND Research Detection System (DS9455) for IF. Briefly, after dewaxing and rehydration, sections were pretreated with the epitope retrieval BOND ER2 high pH buffer (Leica #AR9640) for 20 min at 98°C. Endogenous peroxidase was inactivated with 3% $H_2O_2$ for 10 min at RT. Nonspecific tissue–antibody interactions were blocked by incubating the sections for 30 min at RT with Leica PowerVision IHC/ISH Super Blocking solution (PV6122) for IHC or with the Akoya Biosciences Opal Antibody Diluent/Block solution (ARD1001EA) for IF. The same blocking solution also served as diluent for the primary antibodies. Primary antibodies were incubated on the sections for 45 min at RT. A biotin-free polymeric detection system consisting of HRP conjugated anti-rabbit IgG was then applied for 25 min at RT. For IHC, immunoreactivity was then revealed with the diaminobenzidine (DAB) chromogen reaction. Tissue sections were finally counterstained in hematoxylin, dehydrated in an ethanol series, cleared in xylene, and permanently mounted with a resinous mounting medium (Thermo Scientific ClearVue coverslipper). For IF, the sections were finally incubated with the Akoya Biosciences TSA reagents Opal 520 (OP-1001), 570 (OP-1002), and 690 (OP-1003) (working concentration 1/150) for 10 min at RTs followed by Spectral DAPI nuclear counterstain (Akoya Biosciences FP1490) and mounting with Fluoromount-G (SouthernBiotech 100-01). Negative controls were obtained by replacement of the primary antibodies with irrelevant isotype-matched rabbit antibodies. HE and IHC-stained slides were evaluated by two board-certified veterinary pathologists (ER and CAA) with extensive expertise in mouse pathology. Staging of the seminiferous tubules was performed according to well-established morphological criteria (*Ahmed and de Rooij, 2009*; *Meistrich and Hess, 2013*). The Aperio Versa 200 instrument was used for image

acquisition. Digital image analysis for cell count and morphometry of seminiferous tubules as well as for normalized quantification of marker expression within the SCP-1-positive spermatocyte population was performed using FIJI/ImageJ open-source software (*Schroeder et al., 2021*; *Arena et al., 2017*; *Schindelin et al., 2012*). Values for the normalized quantification correspond to the average positive area per spermatocyte and are expressed in um (*Aitken et al., 2022*).

## Immunoblot analysis

Total testis lysates were prepared by homogenization with a glass-to-glass potter homogenizer on ice in 20 mM HEPES, 100 NaCl, pH 7.4, supplemented with protease and phosphate inhibitors (ROCHE). The lysate was then transferred to a fresh tube, supplemented with Triton- X 1%, SDS 0.1%, and passed several times through a 26-gauge syringe. The samples were then centrifuged at $20,000 \times g$ for 15 min at 4°C to remove insoluble material. Tissue extracts or enriched mitochondrial membranes were separated in reducing and denaturing conditions in NuPage gels (Invitrogen). Proteins were transferred to PVDF 0.45 µm membranes, blocked with milk 5% TRIS-buffered saline, Tween-20 0.1% (TTBS), and incubated with the indicated primary antibodies, washed in TTBS incubated for 1 hr at RT with horseradish peroxidase conjugated secondary antibodies in 5% milk-TTBS or Alexa Fluor conjugated secondary antibodies. Proteins were identified by chemiluminescence or by fluorescence according to the type of secondary antibody. A PARL carboxy-terminal antibody was generated in house as previously reported (*Cipolat et al., 2006*).

## Subcellular fractionation methods

To prepare testis-enriched mitochondrial fractions for western blotting or blue native gel electrophoresis, freshly collected testis was homogenized with a motor-driven Teflon pestle set at 800 rpm in a glass potter containing ice-cold 20 mM HEPES, 225 mM sucrose, 75 mM mannitol, 1 mM EGTA pH 7.4, on ice. For mitochondrial respiration experiments, fresh testis was homogenized manually with a Teflon pestle in ice-cold 20 mM HEPES, 225 mM sucrose, 75 mM mannitol, 1 mM EGTA pH 7.4, on ice, then gently passed through a 22-gauge syringe. The homogenate was centrifuged at $700 \times g$ for 10 min at 4°C to remove nuclei and unbroken debris. The supernatant (tissue homogenate) was then centrifuged at $10,000 \times g$ for 10 min at 4°C to pellet mitochondrial enriched mitochondrial membranes. To prepare liver enriched mitochondrial fractions, freshly collected liver was thoroughly rinsed in homogenization buffer, then homogenized with a motor-driven Teflon pestle set at 800 rpm in a glass potter containing ice-cold 20 mM HEPES, 225 mM sucrose, 75 mM mannitol, 1 mM EGTA pH 7.4, on ice. The homogenate was centrifuged at $1000 \times g$ for 10 min at 4°C to remove nuclei and unbroken debris. The supernatant (tissue homogenate) was then centrifuged at $6000 \times g$ for 10 min at 4°C. Brain mitochondria were purified according to Sims' method (*Sims and Anderson, 2008*).

## Blue native gel electrophoresis

Blue native gel electrophoresis of digitonin-solubilized mitochondria was performed as described (*Jha et al., 2016*). Then, 100 µg isolated mitochondria were solubilized with 600 µg digitonin in Invitrogen Native Page sample buffer on ice for 20 min, then centrifuged at $20,000 \times g$ for 20 min at 4°C. 0.75% Coomassie G-250 was added to supernatants, which were loaded on a 3–12% gradient Invitrogen Native Page gel according to the instructions. After electrophoresis, mitochondrial complexes and super complexes were visualized by protein staining with InstantBlue Coomassie Protein Stain (ISB1L) (Abcam ab119211).

## High-resolution respirometry

Mitochondrial respiration in testis mitochondria respiration was measured in Miro6 Buffer (*Fasching et al., 2016*) (20 mM HEPES, 110 mM sucrose, 10 mM $KH_2PO_4$, 20 mM taurine, 60 mM lactobionic acid, 3 mM $MgCl_2$, 0.5 EGTA, pH 7.1, 1 mg/ml fatty acid-free BSA, catalase 280 U/ml) at 37°C as previously described (*Pesta and Gnaiger, 2012*; *Spinazzi et al., 2019*). When needed $H_2O_2$ was added to reoxygenate the chambers by catalase mediated $O_2$ generation. Then, 150 µg of mitochondrial-enriched membranes were loaded into the Oroboros 2K oxygraph. A typical experiment is illustrated in *Figure 4D*. Oxygen consumption rates were measured before and after addition of the following sequence of substrates and specific inhibitors: (1) 2.5 mM pyruvate, 10 mM glutamate, and 1 mM malate to measure complex I-driven leak respiration (CI leak); (2) 2.5 mM ADP to determine complex

I-driven phosphorylating respiration (CI OXPHOS). (3) 5 mM succinate to determine the phosphory-lating respiration driven by simultaneous activation of complex I and II (CI + II OXPHOS); (4) titrating concentrations of the mitochondrial uncoupler CCCP to reach the maximal uncoupled respiration (CI + II electron transfer capacity, ET); (5) 200 nM rotenone to fully inhibit complex I-driven respiration and measure complex II-driven uncoupled respiration (CII electron transfer capacity, CII ET); (6) 0.5 μM antimycin A to block mitochondrial respiration at the level of complex III. Residual oxygen consump-tion was always negligible. (7) 2 mM ascorbate, 0.5 mM TMPD to measure cytochrome $c$ oxidase (CIV)-driven respiration; (8) 125 μg/ml cytochrome $c$ to evaluate mitochondrial outer membrane integ-rity and (9) 500 μM potassium cyanide (KCN) to specifically block cytochrome $c$ oxidase activity and measure residual background oxygen consumption caused by chemical reaction between ascorbate and TMPD. Cytochrome $c$ oxidase-driven respiration was calculated as the cyanide-sensitive oxygen consumption.

## CoQ analysis
CoQ content and the ratio of the reduced vs. oxidized forms were measured as previously described (*Rodríguez-Aguilera et al., 2017*).

## mtDNA copy number quantification
For mtDNA quantification, total DNA was isolated from 20 to 30 mg of testis tissues by using a DNeasy Blood and tissues kit (QIAGEN). qPCRs were performed in triplicate in 96-well reaction plates (Applied Biosystems). Each reaction (final volume 10 μl) contained 25 ng DNA, 5 μl of Power SYBR-Green PCR Master Mix (Applied Biosystems), and 0.5 μM of each forward and reverse primer. COX1, mitochondrial encoded gene, was amplified and β2 microglobulin (β2 m), nuclear encoded gene, was used as a normalizing control. Fold changes in mtDNA amount were calculated with the ΔΔCt method. The employed primers sequences were Cox1-Mus-F: TTTTCAGGCTTCACCCTAGATGA, Cox1-Mus-R: CCTACGAATATGATGGCGAAGTG, B2m-Mus-F: ATGGGAAGCCGAACATACTG, B2M-Mus-R:CAGTCTCAGTGGGGGTGAAT.

## Electron microscopy
Testes of the indicated genotype were collected and immediately fixed with 2.5% glutaraldehyde, 2% paraformaldehyde in 0.1 M cacodylate buffer pH 7.4. Tissue was stored overnight at 4°C in the fixative solution, washed in 0.1 M cacodylate buffer, and post-fixed for 2 hr at RT with 1% OsO₄, 1.5% $K_4Fe(CN)_6$ in 0.1 M cacodylate buffer. Sections were rinsed, stained with 3% uranyl acetate for 1 hr at 4°C, and dehydrated in graded ethanol concentrations and propyleneoxide, followed by embedding in Epon Resin. Resin blocks were sectioned on a ultramicrotome. Post-staining was performed with 3% uranyl acetate followed by lead citrate staining. Semithin sections were collected on slides and stained with 1% Toluidine blue solution (Sigma-Aldrich). Ultrathin sections (60 nm) were mounted on copper grids and imaged using a JEOL transmission electron microscope.

## Cultured cells
Immortalized mouse embryonic fibroblasts (MEFs) derived from WT and *Parl*⁻/⁻ male mice were cultured in Dulbecco's modified Eagle's medium/F-12 (Gibco) containing 10% fetal bovine serum (Gibco). At 30–40% confluence, the MEFs were transduced using a replication-defective recombinant retroviral expression system (Clontech) with either wild-type (*Parl* WT) or catalytic inactive *Parl S275A* as previ-ously described (*Spinazzi et al., 2019*). Cell lines stably expressing the desired proteins were selected based on their acquired resistance to 5 μg/ml puromycin. Cells were regularly tested to rule out Mycoplasma contamination.

## Statistical analysis
Numerical data are expressed and illustrated in all graph bars as mean ± SD from biological repli-cates. No statistical tests were used to predetermine sample size. Replicates numbers were decided from experience of the techniques performed and practical considerations. Two-sided Student's $t$-test was used to compare differences of all quantitative variables between two groups, and Fisher's exact test was used for the analysis of contingency tables to compare the frequency distribution of

ultrastructural abnormalities in two groups. Significance was calculated using GraphPad. Differences were considered statistically significant for p≤0.05. No data were excluded.

## Acknowledgements

This study was supported by the University of Pennsylvania URF research funding to ER (URF Fall 19-0914) and AFM-Telethon to MS (23019). MS is recipient of an INSERM translational research grant (CIHU INSERM). The authors affiliated with the Penn Vet Comparative Pathology Core are partially subsidized by the Abramson Cancer Center Support Grant (P30 CA016520); the Aperio Versa 200 scanner used for imaging was acquired through an NIH Shared Instrumentation Grant (S10 OD023465-01A1); the Leica BOND RXm instrument used for IHC was acquired through the Penn Vet IIZD Core pilot grant opportunity 2022. We are profoundly grateful to Prof. Bart De Strooper, KU Leuven, for his support and for the generous gift of all mouse strains used in this project. We thank Prof. Jeremy Wang, University of Pennsylvania, for his insightful comments as well as Dr. Cristina Ugalde, University Hospital of Madrid, for her feedback on blue-native electrophoresis results.

## Additional information

### Funding

| Funder | Grant reference number | Author |
|---|---|---|
| University of Pennsylvania | URF Fall 19-0914 | Enrico Radaelli |
| Association Française Myopathies (AFM) Telethon | 23019 | Marco Spinazzi |
| Abramson Cancer Center | P30CA016520 | Enrico Radaelli |

The funders had no role in study design, data collection and interpretation, or the decision to submit the work for publication.

### Author contributions

Enrico Radaelli, Marco Spinazzi, Conceptualization, Resources, Data curation, Formal analysis, Supervision, Funding acquisition, Validation, Investigation, Visualization, Methodology, Writing - original draft, Project administration, Writing - review and editing; Charles-Antoine Assenmacher, Formal analysis, Investigation; Jillian Verrelle, Esha Banerjee, Florence Manero, Salim Khiati, Anais Girona, Investigation; Guillermo Lopez-Lluch, Placido Navas, Investigation, Writing - review and editing

### Author ORCIDs

Enrico Radaelli http://orcid.org/0000-0002-2885-0221
Marco Spinazzi https://orcid.org/0000-0003-0048-9558

### Ethics

Mice were included in a health-monitoring program developed in accordance with guidelines of the Federation of European Laboratory Animal Science Associations (FELASA). All experiments were approved by the Ethical Committee on Animal Experimenting of the University of Leuven (IACUC protocol #072/2015) and by the French Ministry (DUO-OGM 5769 29/3/2019).

### Decision letter and Author response

Decision letter https://doi.org/10.7554/eLife.84710.sa1
Author response https://doi.org/10.7554/eLife.84710.sa2

## Additional files

### Supplementary files

• Transparent reporting form

## Data availability

All data generated or analysed during this study are included in the manuscript and supporting file. Source data files have been included.

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
