## [Editor Report]

This manuscript reports an important finding that spermatogenic defects in Parl KO mice, a genetic model for Leigh syndrome, may result from mitochondrial defects leading to ferroptosis. The finding is of significance because male germ cell ferroptosis has not been well characterized before. The data as a whole strongly support ferroptosis as a mechanism for germ cell death in the Parl KO. However, potential non-ferroptosis and 'accidental' necrosis cannot be excluded, and the potential effects of quantitative immunofluorescent staining, instead of assays using purified spermatogenic cells, on the conclusion drawn should be considered.

---

## [Decision Letter]

**Decision letter after peer review:**

Thank you for submitting your article "Mitochondrial defects leading to arrested spermatogenesis and ferroptosis in a mouse model of Leigh Syndrome" for consideration by *eLife*. Your article has been reviewed by 2 peer reviewers, and the evaluation has been overseen by a Reviewing Editor and Ricardo Azziz as the Senior Editor. The reviewers have opted to remain anonymous.

Essential revisions:

1) Definitive evidence for germ cell ferroptosis.

2) Quantitative analyses using purified spermatogenic cells instead of total testes as the cellular compositions are different between KOs and controls.

*Reviewer #1 (Recommendations for the authors):*

1. In Figure 1A, it would be more useful to show testis/body weight ratio instead of the two metrics separately.

2. In Figure 1, SYCP-1 is used in the associated text but SCP-1 is used in the figure/legend; only one name should be used to avoid confusion.

3. Line 176 typo: "detail" -> "detailed".

4. Figure 2 – supplement 2 (activated caspase staining) is only referenced in association with Figure 6. It seems more appropriate to move this panel to the Figure 6 supplement.

5. For Figure 4B, the method for quantitating mtDNA is not specified in either the text or legend. It is provided in the methods (qPCR) but should also be specified in the main manuscript.

6. The various factors and pathways discussed tested throughout the study are complex. It would be helpful to have a cartoon or model figure outlining the proposed pathway and the differences between WT and Parl-/-.

7. Several details are missing in the figure legends; making the figures difficult to interpret:

a. In the legend to Figure 3A, the protein used as a loading control is not specified.

b. In Figure 4C, it is difficult to know which bands indicate assembly defects compared to normal complex assembly.

c. In Figure 4E, the abbreviation RCR is not explained in either the text or legend.

*Reviewer #2 (Recommendations for the authors):*

1. Ferroptosis is not reported in mammalian germ cells. However, the data presented here only indirectly support the possibility of germ cell ferroptosis in the mutants. The authors should clarify the definitions of ferroptosis in the field and demonstrate definitive evidence of germ-cell ferroptosis in mammals. P53 is not the only marker for ferroptosis. The HNE data is interesting but does not prove the main point.

2. Figure 2—figure supplement 2 shows "Absence of apoptosis in degenerated Parl-/- testis." However, apoptosis needs to be quantitatively evaluated in the mutants. Apoptosis is usually not so frequent in other mutants showing meiotic defects. Also, wild-type mice usually show a low rate of apoptosis. This data is not sufficient to demonstrate the "Absence of apoptosis."

3. The authors have done many analyses using the whole-testis (such as western or others; Figure 3A, 4A, 4B, 4C, 4D, 4E, 5B, 6A, 6B, 6 sup 1A-C, 6 sup 2A). However, the mutant testes were depleted with late germ cells, and the composition of germ cells was apparently different between the wild-type and mutant testes. The authors should confirm these results using analyses of juvenile testes (in which cellular composition is not changed yet) or isolated germ cells of specific stages from wild-type and mutant testes.

4. Results of fertility tests should be provided.

5. Figure 1—figure supplement 1: PARL deficiency should be confirmed in mutant Leydig cells. Otherwise, there is no evidence that PARL is depleted in the conditional mutants, as suggested.

6. Line 121-124: chromosome synapsis was not examined in the mutant. In this context, Line 242-244 explains the chromatin defects in the mutants, but there is no clear characterization.

7. Figure 2—figure supplement 2 should be described in the order of the explanation.

8. Line 155-158: The Pink1 single mutants should be characterized before examining the double mutants. In flies, PINK1 is required for spermatogenesis (Clark et al., Nature 2006: PMID 16672981). Thus, it is interesting if PINK1 is not required for spermatogenesis in mice. Results of fertility tests should be provided here as well.

9. Figure 4D is not possible to interpret. Please show the wild-type and mutant data separately and explain in a way general audiences can understand.

10. The Discussion section is disorganized and hard to read.

[Editors’ note: the authors submitted for reconsideration following the decision after peer review. What follows is the decision letter after the first round of review.]

Thank you for resubmitting your work entitled "Mitochondrial defects leading to arrested spermatogenesis and ferroptosis in the PARL deficient mouse model of Leigh Syndrome" for further consideration by *eLife*. Your revised article has been evaluated by Diane Harper (Senior Editor) and a Reviewing Editor.

The manuscript has been improved but there are some remaining issues that need to be addressed, as outlined below:

1) Add a paragraph summarizing the definitions of ferroptosis in mammals and explain how observation here fulfills these criteria.

2) The cellular compositions are different in KO vs control testes. Specifically, the mutant testes are enriched in spermatogonia, while the controls are enriched in spermatids. Therefore, purification of specific spermatogenic cell types followed by molecular assays is necessary.

*Reviewer #2 (Recommendations for the authors):*

The authors have addressed my primary concerns by providing quantitation of immunofluorescence microscopy and EM images, as well as additional controls and explanations. Functional infertility assessed by co-housing of mutant males with wild type females, which is an important addition to the report of azoospermia in describing an infertility phenotype, is now described at the beginning of the Results section. The paper makes a strong case for ferroptosis as a mechanism for germ cell death in Parl mutants and represents an advance in the fields of male fertility and mitochondrial function. There are two remaining points that I think should be acknowledged in the Discussion section:

1) I agree with reviewer 2 that analysis of stage-specific germ cells is important for the conclusions. In their rebuttal, the authors argue that deletion of Parl in all cells in the germline (whole body) knockout makes this concern irrelevant; this is not true because Parl is likely functioning differently across different germ cell types, making differing cell compositions a potential artifact in bulk assays. In addition, in a whole-body knockout there may be cell non-autonomous effects from testis somatic cells. The data from 4-week testes referred to in the rebuttal will not address the cell composition issue because by 4 weeks (28 days) there will be advanced round spermatids in control testes, meaning that cell compositions will differ between KO and control.

Taking all the data together, I think that the immunofluorescence data strongly supports a cell autonomous effect of Parl knockout in spermatocytes as the authors suggest. However, I think that they should acknowledge the possible issues related to mixed cell populations and somatic cell effects in the Discussion section.

2) Similarly, the data as a whole strongly supports ferroptosis as a mechanism for germ cell death in the Parl KO, but it is difficult to fully exclude non-ferroptotic 'accidental' necrosis. This should also be acknowledged in the Discussion section.

*Reviewer #3 (Recommendations for the authors):*

In this revision, I appreciate the authors' massive efforts to address my previous concerns, but I do not think my main concerns were not effectively addressed. The study provided several indirect evidence of germ cell ferroptosis, but I do not think the results firmly established the occurrence of germ cell ferroptosis. In general, perturbations in mitochondria dynamics could be expected to disrupt spermatogenesis. It would be necessary to clearly define germ-cell ferroptosis to explain the specific phenotype of the PARL mutants. Overall, I appreciate the potential impact; but I am not fully convinced by the main conclusion reported in this study yet.

1. The main issue is that the manuscript, including some of the revised parts, was not clearly written. I still do not understand many parts. Now, the abstract, results, and discussion are hard to read. The authors should clarify the contents and ask a professional editor to clarify the language. I am sorry that it took time to review the revised manuscript.

2. Ferroptosis is not reported in mammalian germ cells. However, the revised manuscript only indirectly supports the possibility of germ-cell ferroptosis in the mutants. The authors did not address my previous concern: the authors should clarify the definitions of ferroptosis in the field and demonstrate definitive evidence of germ-cell ferroptosis in mammals. Again, P53 is not the only marker for ferroptosis. Please add a paragraph summarizing the definitions of ferroptosis in mammals and explain how observation here fulfills these criteria.

3. The mutant testes were depleted with late germ cells, and the composition of germ cells was apparently different between the wild-type and mutant testes. The issues are that the mutant testes are enriched with spermatogonia, and the controls are enriched with the late stages of spermatogenesis. I appreciate the authors have done many analyses using the whole testis in this revision. However, they did not address this main point. I understand it can be challenging to isolate germ cells of specific stages from wild-type and mutant testes. At least, the authors should confirm the main conclusions using analyses of juvenile testes (in which cellular composition is not changed yet). For example, at least the reduction of CoQ (Figure 5A) should be examined to solidify the main conclusion.

4. I suggested that "results of fertility tests should be provided."At least, please show how many mice were examined for what duration.

5. Figure 1—figure supplement 1: PARL deficiency should be confirmed in mutant Leydig cells. In this case, I understand that there is no PARL antibody to confirm. This caveat can be noted.

6. Figure 4D is still not possible to interpret. Please clearly explain what this means in detail. Also, I found a weird mark in the middle of the panel.

7. New Figure 6C: GPX4 expression appears to be reduced in various cells in Parl-/- testes, but I do not see any cell-type specific reduction in spermatocytes. Is GPX4 highly expressed in other stages in the mutants?

---

## [Author Response]

Essential revisions:Reviewer #1 (Recommendations for the authors):1. In Figure 1A, it would be more useful to show testis/body weight ratio instead of the two metrics separately.

We included as suggested the testis/body weight ratio but kept also the information about testis and weight separately to avoid loss of information (the ratio alone for instance could decrease if the KO body weight increased).

2. In Figure 1, SYCP-1 is used in the associated text but SCP-1 is used in the figure/legend; only one name should be used to avoid confusion.

We modify the text as suggested by the reviewer.

3. Line 176 typo: "detail" -> "detailed".

We modify the text as suggested by the reviewer.

4. Figure 2 – supplement 2 (activated caspase staining) is only referenced in association with Figure 6. It seems more appropriate to move this panel to the Figure 6 supplement.

We agree and moved this figure to Figure 6—figure supplement 1 as suggested by the reviewer.

5. For Figure 4B, the method for quantitating mtDNA is not specified in either the text or legend. It is provided in the methods (qPCR) but should also be specified in the main manuscript.

We modify the text as suggested by the reviewer adding this point in the legend. The method is detailed in the methods section.

6. The various factors and pathways discussed tested throughout the study are complex. It would be helpful to have a cartoon or model figure outlining the proposed pathway and the differences between WT and Parl-/-.

We agree with the reviewer. We added this cartoon in Figure 7 as suggested by the reviewer.

7. Several details are missing in the figure legends; making the figures difficult to interpret:a. In the legend to Figure 3A, the protein used as a loading control is not specified.

We specified that HSP60 is the loading control.

b. In Figure 4C, it is difficult to know which bands indicate assembly defects compared to normal complex assembly.

We modify the figure to highlight the identification of the different complexes as suggested by the reviewer.

c. In Figure 4E, the abbreviation RCR is not explained in either the text or legend.

As suggested by the reviewer, we specify in the legend the abbreviation RCR being respiratory control ratio, which is a useful parameter to assess the efficiency of oxidative phosphorylation.

Reviewer #2 (Recommendations for the authors):1. Ferroptosis is not reported in mammalian germ cells. However, the data presented here only indirectly support the possibility of germ cell ferroptosis in the mutants. The authors should clarify the definitions of ferroptosis in the field and demonstrate definitive evidence of germ-cell ferroptosis in mammals. P53 is not the only marker for ferroptosis. The HNE data is interesting but does not prove the main point.

GPX4 is the major suppressors of ferroptosis, and the ablation of this protein alone induces ferroptosis. The virtual absence of GPX4 expression specifically found in *Parl^-/-^* spermatocytes provides in our opinion robust evidence for ferroptosis in our model, since genetic or chemical inactivation of this enzyme alone is sufficient to induce ferroptosis in a variety of models. In addition, we discuss how Coenzyme Q, an established and independent suppressor of ferroptosis, is severely suppressed in *Parl^-/-^* cells providing further evidence for this process.

Moreover, we provided quantitative analysis of 3 additional established biomarkers of ferroptosis in *Parl^-/-^* spermatocytes, confirming highly significant pattern of expression consistent with ferroptosis: HNE (increase), Tfr1 (increase), p53 (increase). Altogether, we believe that our study provides definitive evidence of ferroptosis in spermatocytes. Finally, while this paper was under revision, an independent group led by Thomas Langer published a study on Nature Cell Biology finding increased susceptibility of ferroptosis of *PARL-/-* cells in culture treated with GPX4 inhibitors due to defective Coenzyme Q (Deshwal, S. et al. *Nat Cell Biol* 2023: doi:10.1038/s41556-022-01071-y). We believe that these works, one performed in vivo and the other in vitro, perfectly fit together strengthening both conclusions.

2. Figure 2—figure supplement 2 shows "Absence of apoptosis in degenerated Parl-/- testis." However, apoptosis needs to be quantitatively evaluated in the mutants. Apoptosis is usually not so frequent in other mutants showing meiotic defects. Also, wild-type mice usually show a low rate of apoptosis. This data is not sufficient to demonstrate the "Absence of apoptosis."

We provide now quantitative analysis of apoptosis in Figure 6—figure supplement 1. As suggested by the reviewer, few cells in both genotypes showed caspase 3 activation, so we corrected the sentence on “absence of apoptosis”, which is not accurate. We did not see a quantitative difference in the amount of caspase 3 + cells in the two genotypes. Most importantly, we did not see caspase 3 activation in the degenerating adluminal germ cells strongly indicating that apoptosis was not the main cell death mechanisms. Although we cannot rule out a very subtle participation of apoptosis in addition to ferroptosis in the reported phenotype, we believe we can safely conclude that apoptosis was not the main biological mechanism underlying the massive testis phenotype that we described.

3. The authors have done many analyses using the whole-testis (such as western or others; Figure 3A, 4A, 4B, 4C, 4D, 4E, 5B, 6A, 6B, 6 sup 1A-C, 6 sup 2A). However, the mutant testes were depleted with late germ cells, and the composition of germ cells was apparently different between the wild-type and mutant testes. The authors should confirm these results using analyses of juvenile testes (in which cellular composition is not changed yet) or isolated germ cells of specific stages from wild-type and mutant testes.

We agree with the reviewer that the cellular composition is different in KO vs WT and we added many morphometric and cell quantitative analysis in Figure 1—figure supplement 1 to better address this important point. Indeed, primary spermatocytes and to a lesser extent spermatogonia significantly accumulate in *ParlKO* vs WT due to the complete meiotic block, while there is a complete lack of post-meiotic spermatids. To address the reviewer concerns that changes in cellular composition might have affected our interpretation, we performed a series of cell-specific quantitative analysis in WT and mutant testes in order to eliminate any bias that may originate from differences in cell composition.

Figure 3A: since our study is on a germline KO for PARL, the effect of PARL deficiency on PARL substrates is the same in any cells, as previously reported in Spinazzi et al., 2019, so repeating the experiment on isolated germ cells would not provide any additional insight. Moreover, it is technically unfeasible to perform, being most of the PARL substrates undetectable by IHC due to absence of specific antibodies for this technique.

Figure 4A: as above, the mouse employed in the study is a germline KO for PARL, so PARL is absent in any cell, as previously described (Cipolat S et al. *Cell.* 2006), so repeating the experiment on isolated germ cells would not provide any useful insight. Moreover, there is no specific antibody currently available for PARL IHC.

Figure 4B**:** to address the question of whether mtDNA content may be different in WT vs *Parl^-/-^* spermatocytes, we performed quantitative immunofluorescences experiments with antibodies stained for TFAM, a protein associated with mitochondrial nucleoids commonly used as biomarker for mtDNA abundance, and SCP-1, a marker of primary spermatocytes. We did not observe significant difference of TFAM expression in SCP-1+ cells as shown in the new Figure 4—figure supplement 1. These data rule out the possibility of significant decrease of mtDNA in *Parl^-/-^* spermatocytes, that could explain the drastic mitochondrial respiratory chain defects that we describe.

Figure 4C: isolation of mitochondria, required for blue native gel electrophoresis requires a substantial amount of tissue/cells. This amount is impossible to reach after isolation of specific germ cells. We performed BNGE at an earlier stage (4 weeks), when the amount of tissue is sufficient for mitochondrial isolation, and the results are identical compared to those shown and Figure 4C. We can share this experiment if needed. Moreover, although it is not possible to precisely localize the severity of mitochondrial complex disassembly in different testis cells, we can safely affirm that the abnormalities shown in Figure 4C are definitely pathological since they do not respect the well characterized macromolecular organization of respiratory chain complexes and super complexes that is well known and conserved among different cell types even in different species.

Figure 4D**:** this graph, as explained in the text, is simply an illustrative example to describe to reader the protocol of high-resolution respirometry employed in the study.

Figure 4E: as for 4C, it is not realistic to perform mitochondrial isolation after germ cell isolation for the reasons above specified. Moreover, the procedure of germ cell isolation per se very likely would affect and compromise mitochondrial function and respiration. As explained in the text, to gain cell type-specific insights on mitochondrial function/electron transfer we performed hystoenzymatic assessment of cytochrome c oxydase (COX) activity, shown in Figure 4F. To further confirm our data we performed quantitative immunofluorescence analysis for COX4, a subunit of Complex IV of the respiratory chain, in SCP-1 positive primary spermatocytes, confirming a significant decrease in COX4 expression in primary spermatocytes of *Parl-/-* compared to WT. These results have been included in a new Figure 4—figure supplement 2.

Figure 5B: to address the reviewer concerns we performed quantitative immunofluorescence analysis for COQ4 in SCP-1 positive primary spermatocytes, confirming a significant decrease in COQ4 expression in primary spermatocytes of *Parl-/-* compared to WT. We added this new analysis in a new Figure 5—figure supplement 1.

Figure 6A: to address the reviewer concerns we performed quantitative immunofluorescence analysis for GPX4 in SCP-1 positive spermatocytes, confirming a dramatic decrease in GPX4 expression in primary spermatocytes of *Parl-/-* compared to WT (p=0.0013). We added this new analysis in a new Figure 6—figure supplement 2A. We also evaluated GPX4 expression in Sertoli cells with a similar approach and did not find significant differences (Figure 6—figure supplement 2A).

Figure 6B: to address the reviewer concerns we performed quantitative immunofluorescence analysis for HNE in SCP1 positive primary spermatocytes, confirming a dramatic increase in HNE expression in primary spermatocytes of *Parl-/-* compared to WT (p = 0.0002). We added this new analysis in Figure 6—figure supplement 5B.

Figure 6 sup1A, now Figure 6—figure supplement 3A: this experiment is not performed on whole-testis but on total mouse embryonic fibroblasts, as specified in the figure legend.

Figure 6 sup1B-C, now Figure 6—figure supplement 3B-C: 3B is not performed on whole-testis but on isolated mitochondria from different tissues (1B), and 3C on total brain tissue. These experiments show that the drastic effects on GPX4 and lipid peroxidation are not present in these tissues.

4. Results of fertility tests should be provided.

We specify in the text that the mice are totally infertile due to complete lack of sperm production.

5. Figure 1—figure supplement 1: PARL deficiency should be confirmed in mutant Leydig cells. Otherwise, there is no evidence that PARL is depleted in the conditional mutants, as suggested.

There is no currently available specific antibody for PARL immunohistochemistry, so it is not possible to directly quantify the effect of the Nestin-Cre deletion in Leydig cells at protein level. Nevertheless, the expression of Nestin in Leydig cells, that we and others before us verified, is expected to delete *Parl* by Cre recombinase, as in the nervous system. Moreover our extensive observations indicates that Leydig cells are structurally and functionally unaffected in the germline *Parl-/-* suggesting that Leydig cells are not major players of the drastic germ cell phenotype that we link to PARL deficiency.

6. Line 121-124: chromosome synapsis was not examined in the mutant. In this context, Line 242-244 explains the chromatin defects in the mutants, but there is no clear characterization.

We acknowledge lack of this evidence; however characterization of chromatin defects and chromosome synapsis is not the focus of the paper. We provided a more precise characterization of the meiotic arrest by γH2AX staining that we included in Figure 1—figure supplement 1.

7. Figure 2—figure supplement 2 should be described in the order of the explanation.

We thank the reviewer for pointing this out. We moved this figure to Figure 6—figure supplement 1 as suggested by the reviewer.

8. Line 155-158: The Pink1 single mutants should be characterized before examining the double mutants. In flies, PINK1 is required for spermatogenesis (Clark et al., Nature 2006: PMID 16672981). Thus, it is interesting if PINK1 is not required for spermatogenesis in mice. Results of fertility tests should be provided here as well.

We provided additional histological in data on single *Pink1* and *Pgam5* KO in Figure 3. Moreover, we performed a series of novel AIF1 staining confirming that *Pink1-/-* testis, as well as *Pgam5/-* have normal production of spermatids in sharp contrast with *Parl-/-* which show complete premeiotic maturation arrest. We included these data in Figure 3 – supplement 1. Moreover we specify that *Pink1-/-* mice are fertile as also indicated in the JAX website: https://www.jax.org/strain/017946; in fact *Pink1-/-* mice as well as *Pgam5-/-* and *Ttc19-/-* were also bred as homozygous mutant. Moreover, the phenotype of *PINK1-/-* flies is very different and much more severe than *Pink1-/-* mice which have barely any detectable phenotype and normal lifespan.

9. Figure 4D is not possible to interpret. Please show the wild-type and mutant data separately and explain in a way general audiences can understand.

This figure does not show WT and mutant data altogether so it cannot be split as suggested. It is an illustrative trace of one experiment to graphically illustrate to readers interested in bioenergetics how the high-resolution respirometry was performed, and to reassure of the validity of this delicate experiment. We try to improve this explanation in the text and legend.

10. The Discussion section is disorganized and hard to read.

We thank the reviewer for pointing this out. We substantially rewrote the discussion trying to organize it better and improve readability. To illustrate better the complex pathways that are involved we included a cartoon in Figure 7.

[Editors’ note: the authors submitted for reconsideration following the decision after peer review. What follows is the decision letter after the first round of review.]

The manuscript has been improved but there are some remaining issues that need to be addressed, as outlined below:Reviewer #2 (Recommendations for the authors):The authors have addressed my primary concerns by providing quantitation of immunofluorescence microscopy and EM images, as well as additional controls and explanations. Functional infertility assessed by co-housing of mutant males with wild type females, which is an important addition to the report of azoospermia in describing an infertility phenotype, is now described at the beginning of the Results section. The paper makes a strong case for ferroptosis as a mechanism for germ cell death in Parl mutants and represents an advance in the fields of male fertility and mitochondrial function. There are two remaining points that I think should be acknowledged in the Discussion section:1) I agree with reviewer 2 that analysis of stage-specific germ cells is important for the conclusions. In their rebuttal, the authors argue that deletion of Parl in all cells in the germline (whole body) knockout makes this concern irrelevant; this is not true because Parl is likely functioning differently across different germ cell types, making differing cell compositions a potential artifact in bulk assays. In addition, in a whole-body knockout there may be cell non-autonomous effects from testis somatic cells. The data from 4-week testes referred to in the rebuttal will not address the cell composition issue because by 4 weeks (28 days) there will be advanced round spermatids in control testes, meaning that cell compositions will differ between KO and control.Taking all the data together, I think that the immunofluorescence data strongly supports a cell autonomous effect of Parl knockout in spermatocytes as the authors suggest. However, I think that they should acknowledge the possible issues related to mixed cell populations and somatic cell effects in the Discussion section.

We added this caveat at the end of the first paragraph of the discussion.

2) Similarly, the data as a whole strongly supports ferroptosis as a mechanism for germ cell death in the Parl KO, but it is difficult to fully exclude non-ferroptotic 'accidental' necrosis. This should also be acknowledged in the Discussion section.

We added in the discussion that it is impossible to demonstrate the presence of accidental necrosis in vivo since no specific biomarker is currently available for accidental necrosis. However, whether accidental necrosis contributes in part to the cell death phenotype, this would not change the conclusions of the study.

Reviewer #3 (Recommendations for the authors):In this revision, I appreciate the authors' massive efforts to address my previous concerns, but I do not think my main concerns were not effectively addressed. The study provided several indirect evidence of germ cell ferroptosis, but I do not think the results firmly established the occurrence of germ cell ferroptosis. In general, perturbations in mitochondria dynamics could be expected to disrupt spermatogenesis. It would be necessary to clearly define germ-cell ferroptosis to explain the specific phenotype of the PARL mutants. Overall, I appreciate the potential impact; but I am not fully convinced by the main conclusion reported in this study yet.1. The main issue is that the manuscript, including some of the revised parts, was not clearly written. I still do not understand many parts. Now, the abstract, results, and discussion are hard to read. The authors should clarify the contents and ask a professional editor to clarify the language. I am sorry that it took time to review the revised manuscript.

We acknowledge that paper may be in some parts hard to read given complexity of the topic, and also the massive amount of data that have been added to address the reviewer’s concerns. We have again modified both abstract and discussion with the hope of improving the readability of the content. Upon request of the reviewer, we previously added a cartoon illustrating the key elements of the paper (Figure 7). At this stage, in absence of more precise indications, we cannot further address this point. If the paper is accepted, our English native speaker coauthors will work with the editorial office to perform a thorough style improvement and proofreading before publication.

2. Ferroptosis is not reported in mammalian germ cells. However, the revised manuscript only indirectly supports the possibility of germ-cell ferroptosis in the mutants. The authors did not address my previous concern: the authors should clarify the definitions of ferroptosis in the field and demonstrate definitive evidence of germ-cell ferroptosis in mammals. Again, P53 is not the only marker for ferroptosis. Please add a paragraph summarizing the definitions of ferroptosis in mammals and explain how observation here fulfills these criteria.

In our previous revision we have already addressed this question and explained the definition of ferroptosis and how our observations fulfil these criteria. We highlighted further in this revised version that ferroptosis is a caspase independent type of regulated cell death defined by uncontrolled lipid peroxidation. This is demonstrated by a dramatic increase in 4-HNE signal in degenerating PARL-deficient spermatocytes. Moreover, PARL-deficient spermatocytes have a dramatic cell-specific expression defect of the major ferroptosis inhibitory enzyme GPX4. Genetic or chemical inactivation of this enzyme alone is sufficient to induce ferroptosis in a variety of models both in vitro and in vivo. Finally, we have shown that additional established biomarkers of ferroptosis such Tfr1 and p53 consistently increase in PARL-deficient degenerating spermatocytes, providing unambiguous evidence of ferroptosis in PARL-deficient spermatocytes.

3. The mutant testes were depleted with late germ cells, and the composition of germ cells was apparently different between the wild-type and mutant testes. The issues are that the mutant testes are enriched with spermatogonia, and the controls are enriched with the late stages of spermatogenesis. I appreciate the authors have done many analyses using the whole testis in this revision. However, they did not address this main point. I understand it can be challenging to isolate germ cells of specific stages from wild-type and mutant testes. At least, the authors should confirm the main conclusions using analyses of juvenile testes (in which cellular composition is not changed yet). For example, at least the reduction of CoQ (Figure 5A) should be examined to solidify the main conclusion.

We have already addressed the issue of the different cell type composition in WT and *Parl-/-* testis in the previous revision by repeating all experiments with quantitative immunofluorescence in specific germ cell populations (e.g. SCP-1-positive cells). This alternative approach to germ cell isolation has been deemed appropriate by the Editors to address the issue of different cellular composition between WT and PARL-deficient testis. Furthermore, this method allows us to visualize and quantify the expression of relevant markers within the intact and unperturbed tissue context avoiding the experimental biases associated with the artificial manipulations for isolating germ cells (PMID: 30149006). The quantitative immunofluorescence data have already been included in 7 supplementary figures added to the previous revised version of our manuscript. All these experiments have confirmed and strengthened our original conclusions. Therefore, we believe that purification of germ cells would not add any relevant scientific information. We also think that it would be ethically not acceptable violating the 3Rs rule of animal experimentation.

Repeating the analysis in juvenile testis in which cellular composition is not changed by PARL deficiency is not necessary in our opinion since we already addressed the question of cell type composition as explained above as well as in our previous submission. Moreover, based on preliminary data in our possession, the analysis of earlier time points would not be informative in that context since the cell composition is changed very early on, well before the presence of germ cell degeneration. Ongoing investigations on the early molecular mechanisms underlying the PARL-deficient phenotype will be part of an independent study which goes beyond the scope of this publication.

4. I suggested that "results of fertility tests should be provided."At least, please show how many mice were examined for what duration.

We have checked this more carefully and concluded that it is impossible to provide meaningful information on this point, since mice acquired full fertility after 6-8 weeks of life but *Parl-/-* mice develop neurological abnormalities by the age of 6 weeks and die by the age of 7 weeks. Therefore, although we know for sure that *Parl-/-* mice are not able to fecundate WT females, we cannot use this argument to. Therefore, we deleted this sentence from the manuscript. The same holds true for PARL double and triple KO (Parl-/-/Pink1-/-); Parl and Pgam5 (Parl-/-/Pgam5-/-); Pink1 and Pgam5 (Pink1-/-/Pgam5-/-); and Parl, Pink1, and Pgam5 combined (Parl-/-/Pink1-/-/Pgam5-/-). We modified Figure 3 accordingly.

Nevertheless, our data clearly show that *Parl-/-* mice are indisputably sterile due to total lack of sperm production caused by completely arrested spermatogenesis and consequent azoospermia. This has been clearly documented through our detailed histological analysis and AIF1 staining, as specified in the previous revision. In conclusion, we believe that fertility tests are superfluous since we have demonstrated that no spermatozoa are produced in any PARL deficient mouse line, which are therefore necessarily infertile.

5. Figure 1—figure supplement 1: PARL deficiency should be confirmed in mutant Leydig cells. In this case, I understand that there is no PARL antibody to confirm. This caveat can be noted.

We specified this caveat in the manuscript.

6. Figure 4D is still not possible to interpret. Please clearly explain what this means in detail.

We are confused by this repeated request, since a very detailed explanation has already been specified in the legend, in the text, and in the rebuttal letter of the previous submission. This figure is an illustrative trace, as provided by the Oroboros 2K high resolution respirometer, of a high resolution respirometry protocol that has been used in the study. The Oroboros 2K respirometer is currently the state-of-the-art instrument to perform the oxygen consumption analysis. This should be especially interesting for scientists interested in mitochondrial bioenergetics since we are not aware of previous studies/methods to perform high-resolution respirometry in testis mitochondria. We believe it is important to publish at least one illustrative trace of similar experiments in order to explain visually the experiment and build solid confidence in the results. We believe that a full course on high-resolution respirometry is out of scope in the paper. For further informations on high resolution respirometry it is possible to find extensive literature elsewhere (some examples: PMID: 18536644, PMID: 27008969, PMID: 32200800) and in the BIOBLAST website https://www.bioblast.at/index.php/MitoPedia:_SUIT

Also, I found a weird mark in the middle of the panel.

We erased the Oroboros 2k symbol that was automatically attached by the software DATLAB.

7. New Figure 6C: GPX4 expression appears to be reduced in various cells in Parl-/- testes, but I do not see any cell-type specific reduction in spermatocytes. Is GPX4 highly expressed in other stages in the mutants?

We are puzzled by this comment. As explained in the text, Fig6C and its insets show exactly the opposite: a clearly reduced expression of GPX4 expression in *Parl-/-* spermatocytes but normal expression in Leydig cells. To corroborate these finding, we added in the previous submission ad hoc experiments with quantitative immunofluorescence showing a dramatic reduction of GPX4 in SCP-1-positive spermatocytes but not in Sertoli cells.